# Inducible and reversible inhibition of miRNA-mediated gene repression in vivo

Gaspare La Rocca[1]*[†], Bryan King[1†], Bing Shui[2], Xiaoyi Li[1,3], Minsi Zhang[1], Kemal M Akat[4], Paul Ogrodowski[1], Chiara Mastroleo[1], Kevin Chen[1], Vincenzo Cavalieri[5], Yilun Ma[6], Viviana Anelli[7], Doron Betel[8], Joana Vidigal[9], Thomas Tuschl[4], Gunter Meister[10], Craig B Thompson[1], Tullia Lindsten[11], Kevin Haigis[2], Andrea Ventura[1]*

[1]Cancer Biology and Genetics Program, Memorial Sloan Kettering Cancer Center, New York, United States; [2]Department of Cancer Biology, Dana Farber Cancer Institute, Boston, United States; [3]Louis V. Gerstner Jr. Graduate School of Biomedical Sciences, Memorial Sloan Kettering Cancer Center, New York, United States; [4]Laboratory of RNA Molecular Biology, The Rockefeller University, New York, United States; [5]Department of Biological, Chemical and Pharmaceutical Sciences and Technologies, University of Palermo, Palermo, Italy; [6]Weill Cornell/Rockefeller/Sloan-Kettering Tri-Institutional MD-PhD Program, New York, United States; [7]Center of Integrative Biology, University of Trento, Trento, Italy; [8]Hem/Oncology, Medicine and Institution for Computational Biomedicine, Weill Cornell Medical College, New York, United States; [9]Laboratory of Biochemistry and Molecular Biology, National Cancer Institute, Bethesda, United States; [10]Regensburg Center for Biochemistry, University of Regensburg, Regensburg, Germany; [11]Immunology Program, Memorial Sloan Kettering Cancer Center, New York, United States

*For correspondence:
laroccag@mskcc.org (GLR);
venturaa@mskcc.org (AV)

[†]These authors contributed equally to this work

**Abstract** Although virtually all gene networks are predicted to be controlled by miRNAs, the contribution of this important layer of gene regulation to tissue homeostasis in adult animals remains unclear. Gain and loss-of-function experiments have provided key insights into the specific function of individual miRNAs, but effective genetic tools to study the functional consequences of global inhibition of miRNA activity in vivo are lacking. Here we report the generation and characterization of a genetically engineered mouse strain in which miRNA-mediated gene repression can be reversibly inhibited without affecting miRNA biogenesis or abundance. We demonstrate the usefulness of this strategy by investigating the consequences of acute inhibition of miRNA function in adult animals. We find that different tissues and organs respond differently to global loss of miRNA function. While miRNA-mediated gene repression is essential for the homeostasis of the heart and the skeletal muscle, it is largely dispensable in the majority of other organs. Even in tissues where it is not required for homeostasis, such as the intestine and hematopoietic system, miRNA activity can become essential during regeneration following acute injury. These data support a model where many metazoan tissues primarily rely on miRNA function to respond to potentially pathogenic events.

## Introduction

MicroRNAs (miRNAs) are short non-coding RNAs that in Metazoa repress gene expression at the post-transcriptional level by binding to partially complementary sequences on target mRNAs (*Bartel, 2009*; *Bartel, 2018*; *Eichhorn et al., 2014*; *Izaurralde, 2015*).

miRNAs act as part of a large ribonucleoprotein complex known as the miRNA-induced silencing complex (miRISC). In mammals, the Argonaute protein family (AGO1-4) and the trinucleotide repeat-containing gene 6 protein family (TNRC6A/GW182, TNRC6B and TNRC6C) are the core components of the miRISC. AGO binds to the miRNA and facilitates its interaction with target mRNAs (*Schirle et al., 2014*). In turn, TNRC6 binds to AGO and recruits the decapping and deadenylation complexes, leading to degradation of target mRNAs (*Braun et al., 2011*; *Chekulaeva et al., 2011*; *Chen et al., 2009*; *Chen et al., 2014*; *Fabian et al., 2011*; *Guo et al., 2010a*; *Huntzinger et al., 2013*; *Lazzaretti et al., 2009*; *Nishihara et al., 2013*; *Rehwinkel et al., 2005*; *Till et al., 2007*).

Although miRNAs are abundantly expressed in embryonic and adult mouse tissues, and computational and experimental analyses indicate that they target components of virtually every cellular process (*Flynt and Lai, 2008*), animals harboring targeted deletion of single miRNA genes are often indistinguishable from their wild-type counterparts (*Abdellatif, 2012*; *Chivukula et al., 2014*; *Cimmino et al., 2005*; *Liu et al., 2008*; *Park et al., 2010*; *van Rooij et al., 2007*; *Vechetti et al., 2019*; *Williams et al., 2009*). One explanation for these observations is that the redundant functions of related miRNAs may buffer the emergence of obvious phenotypes in mutant animals (*Bartel, 2009*; *Bartel, 2018*). Interestingly, however, clear phenotypes often emerge in mutant adult animals when exposed to external or internal perturbations (*Chivukula et al., 2014*; *Mendell and Olson, 2012*; *van Rooij et al., 2007*). These observations suggest that, at least in some contexts, miRNA function is conditionally, rather than constitutively, required to carry on cellular processes.

Previous efforts to investigate the consequences of global inhibition of miRNA function have relied upon the targeted deletion of the core miRNA biogenesis factors DICER, DROSHA, and DGCR8 (*Treiber et al., 2019*). Several animal models harboring conditional or constitutive knockout alleles of these genes have been generated (*Bernstein et al., 2003*; *Chong et al., 2008*; *Hebert et al., 2010*; *Huang et al., 2012*; *JnBaptiste et al., 2017*; *Kanellopoulou et al., 2005*; *Kobayashi et al., 2015*; *Kumar et al., 2007*; *Wang et al., 2007*). Although these strategies have provided important insights into miRNA biology, they suffer from several limitations.

First, inactivation of these gene products is known to have other consequences in addition to impairing miRNA biogenesis. For instance, DICER is involved in epigenetic regulation in the nucleus in a miRNA-independent manner (*Fukagawa et al., 2004*; *Giles et al., 2010*; *Gullerova and Proudfoot, 2012*; *Okamura and Lai, 2008*; *Song and Rossi, 2017*; *Tam et al., 2008*) and is essential to metabolize transcripts from short interspersed nuclear elements, predominantly Alu RNAs in humans and B1 and B2 RNAs in rodents (*Kaneko et al., 2011*). DROSHA, on the other hand, regulates the expression of several coding and non-coding RNAs by directly cleaving stem–loop structures embedded within the transcripts (*Chong et al., 2010*). Furthermore, DICER and DROSHA are also involved in the DNA-damage response (*Francia et al., 2012*; *Michelini et al., 2017*), and DGCR8 regulates the maturation of small nucleolar RNAs and of some long non-coding RNAs (*Cirera-Salinas et al., 2017*; *Macias et al., 2015*). Consequently, the phenotypes observed in these models cannot be solely attributed to inhibition of miRNA activity.

Another limitation of conditional ablation of miRNA biogenesis genes in vivo is that due to their high stability mature miRNAs can persist for several days after their biogenesis is inhibited. For example, 4 weeks after near complete conditional ablation of *Dicer1* in the muscle, the levels of the highest expressed miRNAs were found to be only reduced by 30–40% and their expression remained substantial even 18 months later (*Vechetti et al., 2019*). This complicates the interpretation of experiments based on temporally controlled conditional ablation of these biogenesis factors, especially in non-proliferating tissues.

Third, a subset of mammalian miRNAs does not rely on the canonical biosynthesis pathway, and therefore their expression and activity are not affected by inactivation of the core miRNA biogenesis factors (*Cheloufi et al., 2010*; *Chong et al., 2010*; *Cifuentes et al., 2010*; *Kim et al., 2016*; *Okamura et al., 2007*; *Ruby et al., 2007*; *Yang and Lai, 2011*).

Finally, these genetic approaches are not reversible and therefore these animal models cannot be used to study the effects of transient inhibition of miRNA function.

To circumvent these limitations, we have generated a novel genetically engineered mouse strain that allows inducible and reversible disassembly of the miRISC, thereby achieving controllable inhibition of miRNA-mediated gene repression in vivo without affecting small RNA biogenesis. To address the reliance of adult tissues on miRNA-mediated gene repression, we have used this novel strain to

investigate the consequences of acute inhibition of the miRISC under homeostatic conditions, and during tissue regeneration.

## Results

## Inhibition of the miRNA pathway through peptide-mediated disruption of the miRISC

Multiple motifs within the N-terminal domain of TNRC6 proteins contain regularly spaced tryptophan residues that mediate the interaction between AGO and TNRC6 by inserting into conserved hydrophobic pockets located on AGO's Piwi domain (*Lian et al., 2009*; *Sheu-Gruttadauria and MacRae, 2018*).

A peptide encompassing one of the AGO-interacting motifs of human TNRC6B has been previously employed as an alternative to antibody-based approaches to efficiently pull down all AGO family members from cell and tissue extracts (*Hauptmann et al., 2015*; *Pfaff et al., 2013*). This peptide, named T6B, competes with endogenous TNRC6 proteins for binding to AGOs. However, as it lacks the domains necessary for the recruitment of decapping and deadenylation factors, it prevents the assembly of the full miRISC, thus resulting in effective inhibition of miRISC-mediated gene repression in cells (*Danner et al., 2017*; *Hauptmann et al., 2015*).

Based on these results, we reasoned that temporally and spatially controlled expression of a T6B transgene in animals would offer the unprecedented opportunity to study the consequences of acute and reversible inhibition of miRNA function in vivo without interfering with miRNA biogenesis or abundance (*Figure 1A*).

To test the suitability of this approach, we first investigated the dynamics of interaction between T6B and the miRISC in mouse and human cell lines. We employed a previously reported size-exclusion chromatography (SEC)-based assay (*La Rocca et al., 2015*; *Olejniczak et al., 2013*) to analyze the molecular weight of AGO-containing complexes in lysates from cells expressing either a doxycycline-inducible FLAG-HA-T6B-YFP fusion protein (hereafter referred to as T6B) or a mutant version (hereafter referred to as T6B$^{Mut}$) incapable of binding to AGO (*Figure 1—figure supplement 1*). We reasoned that if T6B expression prevents AGO from stably binding to TNRC6 and its targets, AGO proteins should be detected in fractions corresponding to ~120–130 kDa, the sum of the molecular weights of AGO (~95 kDa) and the T6B fusion protein (~30 kDa). In contrast, unperturbed AGO complexes that are part of the fully assembled miRISC bound to mRNAs should elute in the void of the column, which contains complexes larger than 2 MDa (*Figure 1B*).

As expected, in lysates from cells expressing no T6B or T6B$^{Mut}$, AGO2 and TNRC6A were mostly detected in the high-molecular-weight fractions, indicating the presence of target-bound miRISC (*Figure 1C*). In contrast, AGO2 and TNRC6A were nearly completely depleted from the high-molecular-weight fractions in lysates from cells expressing T6B (*Figure 1C*). Moreover, while AGO2, TNRC6A, and the polyA-binding protein 1 (PABP1) co-fractionated in lysates from control cells, they eluted in different fractions in lysates from T6B-expressing cells (*Figure 1C*), indicating that T6B leads to loss of interactions between the miRISC components and mRNAs. As expected based on the strong evolutionary conservation of human and mouse AGO and TNRC6 proteins (*Pfaff et al., 2013*; *Zielezinski and Karlowski, 2015*; *Zipprich et al., 2009*), we obtained identical results when human T6B was expressed in mouse embryo fibroblasts (MEFs; *Figure 1—figure supplement 2*).

To test whether the redistribution of AGO-containing complexes induced by T6B expression was mirrored by a loss of miRNA-mediated gene repression, we performed RNAseq analysis on MEFs expressing T6B or T6B$^{Mut}$. Cells expressing T6B displayed marked and selective de-repression of predicted mRNA targets for expressed miRNAs (*Figure 1D*). The extent of de-repression was roughly proportional to the abundance of individual miRNA families, with predicted targets of poorly expressed miRNAs collectively showing modest de-repression compared to targets of more abundantly expressed miRNA families (*Figure 1D*). Importantly, de-repression of miRNA targets was not accompanied by a global change in mature miRNAs levels (*Figure 1E*), consistent with the role of T6B in perturbing the effector step of the miRNA pathway, without affecting miRNA processing.

Of the four mammalian AGO proteins, AGO2 is the only one that has endo-ribonucleolytic activity, which does not require TNRC6 (*Liu et al., 2019*) and is triggered when the AGO2-loaded small RNA and the target are perfectly complementary (*Doench et al., 2003*; *Liu et al., 2004*; *Zeng et al.,*

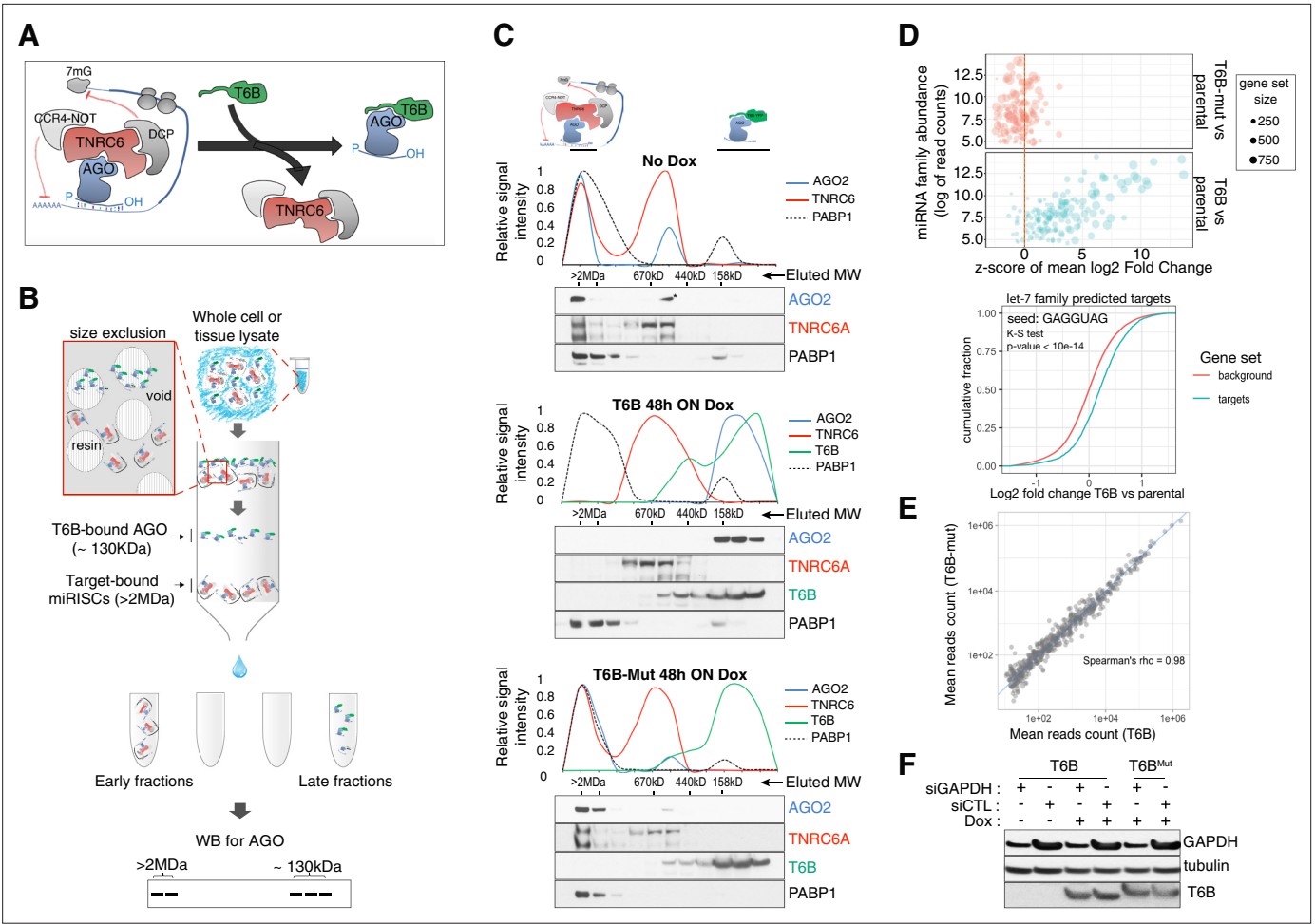

**Figure 1.** T6B fusion protein prevents miRNA-induced silencing complex (miRISC) assembly and impairs microRNA (miRNA) activity in vitro. (**A**) Schematics of T6B action: T6B competes with TNRC6 for binding to AGO proteins preventing miRISC assembly. (**B**) Schematics of the size-exclusion chromatography (SEC) assay for the fractionation of AGO-containing complexes according to their molecular weight. (**C**) SEC profiling of miRISC components upon T6B expression: total lysates from HCT116 cells expressing no fusion protein (upper panel), T6B (middle panel), or T6B[Mut] (lower panel) were fractionated as described in (**B**) and immunoblotted to detect AGO2, TNRC6A, T6B, and PABP1. For each blot, the relative signal intensity was assessed by densitometric analysis. (**D**) RNAseq analysis of total and small RNAs isolated from mouse embryo fibroblasts (MEFs) cell lines expressing either no fusion protein, T6B, or T6B[Mut] (n = 3 for each cell line). Upper panel: bubble plot of target de-repression against miRNA abundance. The mean log2-fold change (T6B or T6B[Mut] vs. control) of predicted targets for each conserved miRNA family was calculated, converted to a z-score and is plotted on the x-axis against the miRNA family abundance (log of the sum of read counts for each member of the family). The size of each circle is proportional to the number of predicted targets. A positive z-score indicates that the targets for that family are preferentially upregulated upon T6B expression, while a negative score would indicate preferential downregulation. Expression of T6B, but not of T6B[Mut], causes preferential upregulation of miRNA targets of the most miRNA families and the effect is roughly proportional to each miRNA family abundance. Lower panel: cumulative distribution plot of predicted let-7 targets compared to background in T6B-expressing MEFs. (**E**) Scatter plots of miRNA abundance as determined by small-RNAseq of total RNA extracted from MEFs expressing either T6B or T6B[Mut] (n = 3 for each cell line). Each dot represents a miRNA in miRbase. (**F**) Effect of T6B expression on AGO2 slicing activity. MEFs expressing either T6B or T6B[Mut] were transfected with siRNAs targeting GAPDH mRNA (siGAPDH) or with scramble siRNA (siCTL). Levels of GAPDH, T6B, and tubulin were assessed by immunoblot 72 hr post-transfection. T6B and T6B[Mut] have slightly different migration on PAGE, as previously observed by *Hauptmann et al., 2015*.

The online version of this article includes the following figure supplement(s) for figure 1:

**Source data 1.** RNAseq, differential gene expression, mouse embryo fibroblasts.

**Source data 2.** z-scores and miRNA family abundance, mouse embryo fibroblasts.

**Source data 3.** Small RNAseq, microRNA counts, mouse embryo fibroblasts.

**Source data 4.** Unedited blots shown in *Figure 1C*.

**Source data 5.** Unedited blots shown in *Figure 1F*.

**Source data 6.** Unedited blots shown in *Figure 1F*.

*Figure 1 continued on next page*

*Figure 1 continued*

**Source data 7.** Uncropped blots shown in *Figure 1F*.

**Figure supplement 1.** Sequence and properties of the FH-T6B-YFP fusion protein.

**Figure supplement 1—source data 1.** Unedited blots shown in *Figure 1—figure supplement 1*.

**Figure supplement 1—source data 2.** Uncropped blots shown in *Figure 1—figure supplement 1*.

**Figure supplement 2.** Size-exclusion chromatography was performed on whole-cell lysates from mouse embryo fibroblasts transduced with retroviral vectors expressing a doxycycline-inducible T6B or T6B^Mut transgene and cultured in the presence of doxycycline for 48 hr.

**Figure supplement 2—source data 1.** Unedited blots shown in *Figure 1—figure supplement 2*.

**Figure supplement 2—source data 2.** Uncropped blots shown in *Figure 1—figure supplement 2*.

*2003*). AGO2's catalytic activity is essential for gene regulation in the germline. For example, in mouse oocytes, AGO2 loaded with endogenous small-interfering RNAs (endo-siRNAs) mediates the cleavage of coding and non-coding transcripts bearing perfectly complementary sequences (*Stein et al., 2015*; *Tam et al., 2008*). In metazoan somatic tissues, in contrast, AGO2 catalytic activity is mainly involved in the biogenesis of miR-486 and miR-451 in the hematopoietic system (*Cheloufi et al., 2010*; *Jee et al., 2018*), and in occasional instances of miRNA-directed cleavage of mRNAs (*Yekta et al., 2004*).

Importantly, T6B expression does not interfere with the ability of synthetic siRNAs to cleave perfectly complementary endogenous targets (*Figure 1F*), indicating that AGO2's catalytic function is not affected by the binding of T6B, and implying that the loading of small RNAs onto AGOs is also not perturbed by T6B.

Collectively these results demonstrate that ectopic T6B expression in mammalian cells causes global inhibition of miRISC function with minimal perturbation of the expression of mature miRNAs, and with preservation of AGO2's endo-nucleolytic activity.

## Generation of a mouse strain with inducible expression of a T6B transgene

To apply this general strategy to an in vivo setting, we next generated mouse embryonic stem cells (mESCs) expressing a doxycycline-inducible T6B transgene. We used a knock-in approach in which the doxycycline-inducible transgene is inserted into the *Col1a1* locus of mESC expressing the reverse tetracycline-controlled transactivator (rtTA) under the control of the endogenous *Rosa26* promoter (*Beard et al., 2006*; *Figure 2A*). Targeted mESCs were tested for the ability to express the T6B transgene in response to doxycycline (*Figure 2—figure supplement 1*) and then used to generate mice with genotype *Rosa26^rtTA/rtTA*; *Col1a1^T6B/T6B* (hereafter R26^T6B). *Rosa26^rtTA/rtTA*; *Col1a1^+/+* mice, with untargeted *Col1a1* loci but expressing rtTA served as negative controls (hereafter R26^CTL).

Upon doxycycline administration, we observed strong expression of T6B in R26^T6B mice and across most adult tissues (*Figure 2B*). Notable exceptions were the central nervous system (*Figure 2B*, *Figure 2—figure supplement 2*), probably due to low blood–brain barrier penetration of doxycycline, and the skeletal muscle and the heart, most likely due to low expression of the rtTA transgene in these tissues (*Premsrirut et al., 2011*).

When doxycycline was administered in the diet, T6B became detectable after 24 hr, reached a plateau after 3 days, and completely disappeared 4 days after doxycycline removal from the diet (*Figure 2C*).

Because colon and liver expressed uniformly high levels of T6B in response to doxycycline, we used these tissues to test the effects of T6B expression on miRISC activity in vivo. Co-IP experiments using antibodies directed to T6B confirmed the interaction between AGO and T6B in these tissues (*Figure 2D*, *Figure 2—figure supplement 3*). Expression of T6B resulted in nearly complete disassembly of the miRISC, as indicated by the elution shift of AGO from the high-molecular-weight to low-molecular-weight fractions in both tissues (*Figure 2E*, *Figure 2—figure supplement 4*). Importantly, doxycycline removal from the diet led to a complete reconstitution of the miRISC, as indicated by the reappearance of AGO2 in the high-molecular-weight fractions (*Figure 2E*).

To test whether T6B expression also resulted in inhibition of miRNA-mediated gene repression in vivo, we performed RNAseq on total RNAs extracted from the liver and colon of R26^T6B and R26^CTL mice kept on doxycycline-containing diet for 1 week. As shown in *Figure 2F*, T6B expression resulted in marked de-repression of miRNA targets in both tissues.

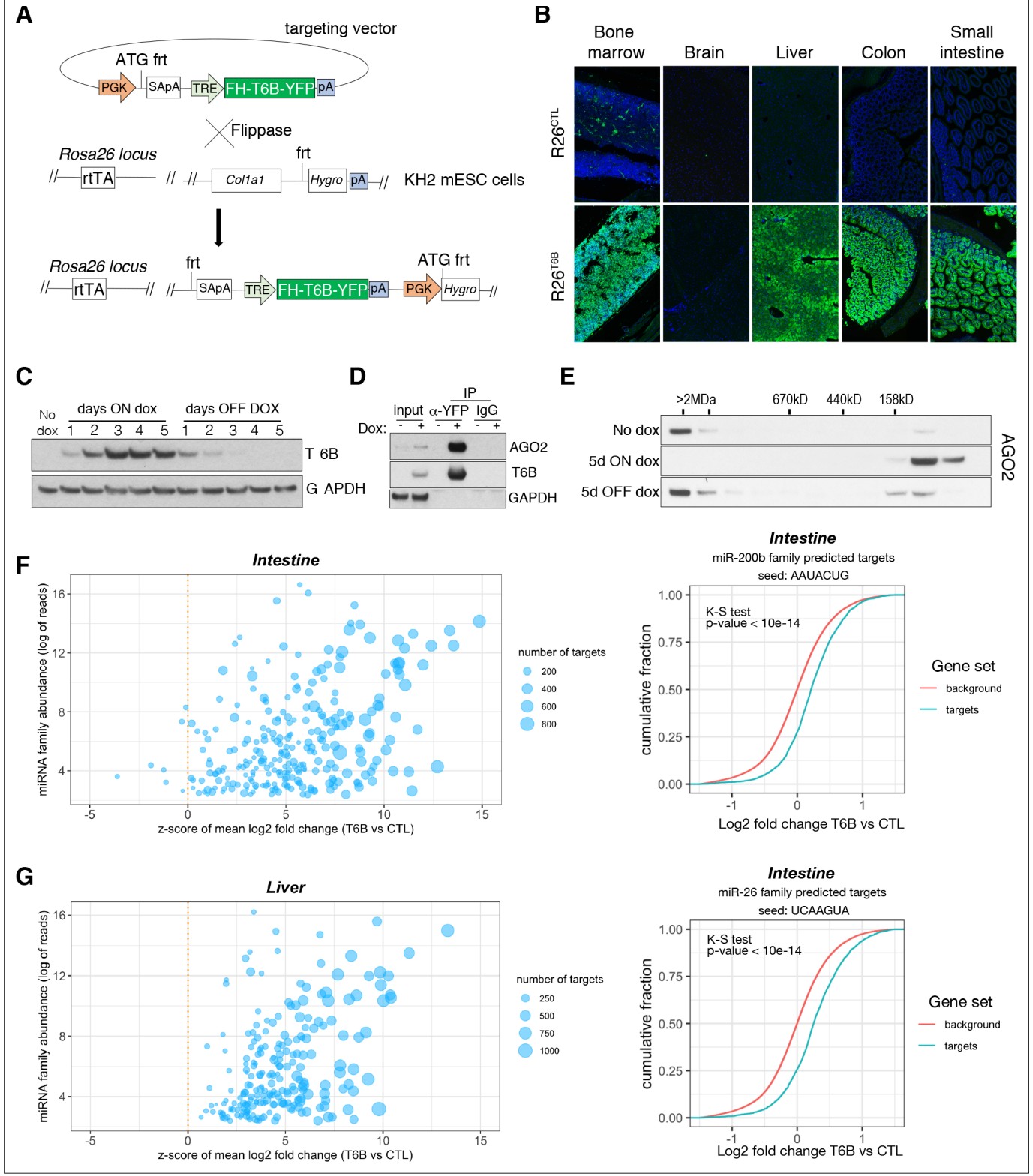

**Figure 2.** Expression of T6B reversibly blocks miRNA-induced silencing complex (miRISC) assembly and inhibits microRNA (miRNA) function in vivo. (**A**) Schematic of the targeting strategy to generate the T6B mouse. The construct contains a flippase recognition target site (frt) that allows homing into the *Col1a1* locus when electroporated together with a vector expressing the Flippase recombinase into KH2 (*Col1a1-frt/Rosa26-rtTA*) murine embryonic stem cells. KH2 also express the rtTA trans-activator driven by the endogenous *Rosa26* (R26) promoter. (**B**) Immunofluorescence imaging performed using an anti-YFP antibody, showing T6B expression in a panel of tissues of adult R26$^{T6B}$ mice fed doxycycline for 7 days. Tissues from R26$^{CTL}$ (carrying

*Figure 2 continued on next page*

*Figure 2 continued*

the rtTA allele but not the T6B allele) were used as negative controls. (**C**) Protein lysates from the liver of R26$^{T6B}$ mice on or off doxycycline-containing chow for the indicated number of days were resolved by SDS-PAGE and western blotting was performed with anti-HA antibody to detect expression of the T6B transgene. (**D**) Co-IP experiments using an anti-YFP antibody showing interaction between AGO and T6B in total liver extracts from T6B mice on doxycycline-containing chow. (**E**) Size-exclusion chromatography (SEC) elution profile of AGO2-containing complexes in liver lysates from T6B mice euthanized at the indicated time points after doxycycline administration. Notice the shift of AGO2 from the high-molecular-weight fractions to the low-molecular-weight fractions after 5 days of doxycycline treatment and the reconstitution of the full miRISC after removal of doxycycline from the diet. (**F, G**) Total RNA extracted from the large intestine (**F**) and the liver (**G**) of R26$^{CTL}$ and R26$^{T6B}$ mice was subjected to RNAseq (n = 3 for each strain). Left panel: scatter plot showing the effect of T6B expression on targets of all miRNA families was generated as described in *Figure 1D*. The abundance of each miRNA family was calculated using dataset from *Isakova et al., 2020*. Right panel: representative cumulative distribution plot of log2-fold changes in expression of predicted targets of the indicated miRNA families.

The online version of this article includes the following figure supplement(s) for figure 2:

**Source data 1.** RNAseq, differential gene expression, colon and liver.

**Source data 2.** Z-scores and miRNA families abundance, colon and liver.

**Source data 3.** Unedited blots shown in *Figure 2C*.

**Source data 4.** Uncropped blots shown in *Figure 2C*.

**Source data 5.** Unedited blots shown in *Figure 2D*.

**Source data 6.** Uncropped blots shown in *Figure 2D*.

**Source data 7.** Unedited blots shown in *Figure 2E*.

**Source data 8.** Uncropped blots shown in *Figure 2E*.

**Figure supplement 1.** Expression of FH-T6B-YFP fusion protein in targeted ES clones.

**Figure supplement 1—source data 1.** Unedited blots shown in *Figure 2—figure supplement 1*.

**Figure supplement 1—source data 2.** Uncropped blots shown in *Figure 2—figure supplement 1*.

**Figure supplement 2.** Immunofluorescence imaging using a YFP-specific antibody, showing T6B expression in a panel of tissues of adult R26$^{T6B}$ mice (second column) and CAG$^{T6B}$ mice (third column) fed doxycycline-containing diet for 7 days.

**Figure supplement 3.** Total extracts from the colon of R26$^{T6B}$ mice kept on doxycycline-containing diet for 1 week were immunoprecipitated using an anti-YFP antibody and probed with the indicated antibodies to measure the interaction between the T6B fusion protein and AGO2 in vivo.

**Figure supplement 3—source data 1.** Unedited blots shown in *Figure 2—figure supplement 3*.

**Figure supplement 3—source data 2.** Uncropped blots shown in *Figure 2—figure supplement 3*.

**Figure supplement 4.** Size-exclusion chromatography (SEC) fractionation followed by western blotting of total extracts from the liver and large intestine of control and R26$^{T6B}$ mice treated with doxycycline-containing chow for 7 days.

**Figure supplement 4—source data 1.** Unedited blots shown in *Figure 2—figure supplement 4*.

**Figure supplement 4—source data 2.** Uncropped blots shown in *Figure 2—figure supplement 4*.

Based on these results, we conclude that T6B expression allows acute and reversible disruption of the miRISC, and concomitant inhibition of miRNA function in vivo.

## Consequences of miRISC disruption in adult tissues under homeostatic conditions

Given the central role of miRNAs in gene regulatory networks, one might expect widespread phenotypes emerging when miRISC function is systemically inhibited. Consistent with this hypothesis, inhibition of miRISC starting either at conception (*Figure 3A*) or at mid-gestation caused developmental defects and perinatal lethality in R26$^{T6B}$ mice (*Figure 3B*, *Figure 3—figure supplement 1*). Histological examination of hematoxylin-eosin-stained sections of P0 R26$^{T6B}$ pups treated with doxycycline starting at mid-gestation confirmed a general delay in development and reduced growth, but no specific organ defects. Surprisingly, however, adult R26$^{T6B}$ mice kept on doxycycline diet for up to 2 months remained healthy and appeared normal upon macroscopic and histopathological examination.

Detailed examination of the intestine confirmed extensive T6B expression in the epithelium and in the mesenchymal compartment (*Figure 3—figure supplement 2*), but no architectural abnormalities were observed (*Figure 3C*). Cells in the crypts showed no significant changes in expression pattern of Ki67 protein (*Figure 3—figure supplement 3*), suggesting that the proliferation and turnover of

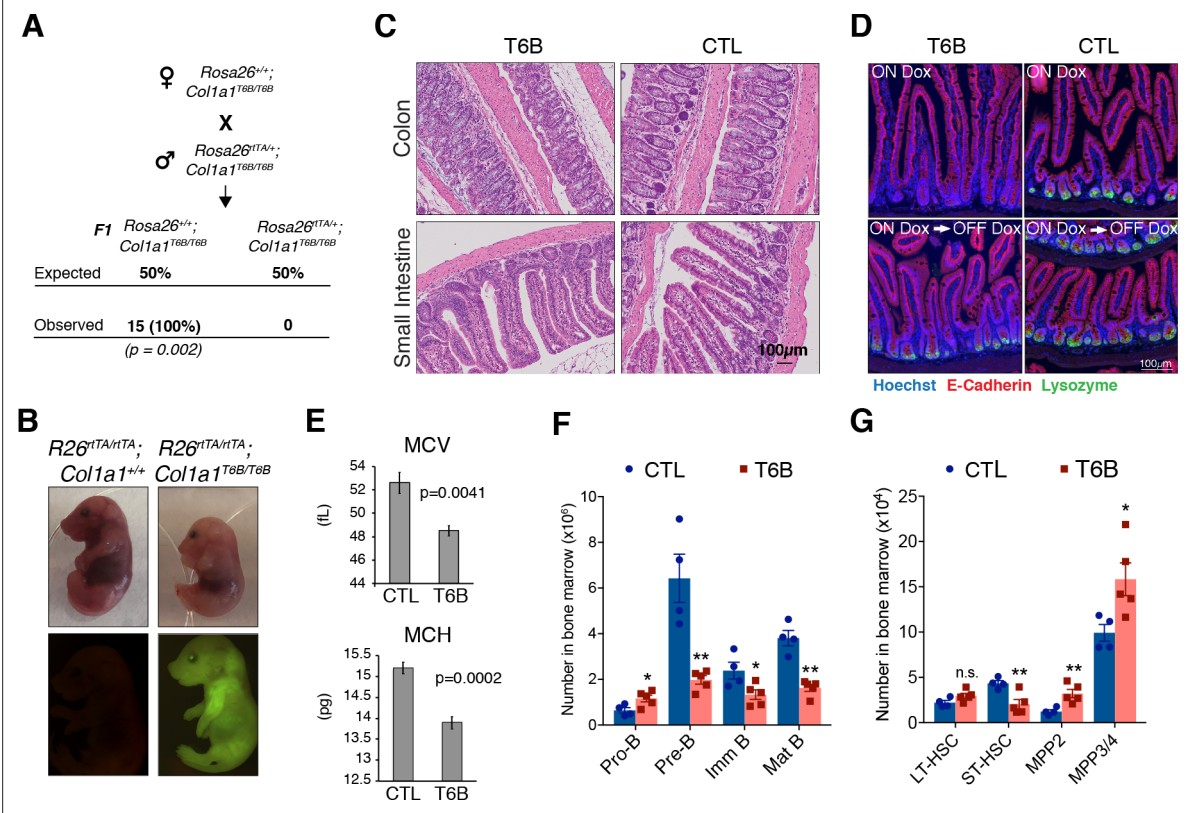

**Figure 3.** Phenotypic analysis of R26<sup>T6B</sup> mice during homeostasis. (**A**) *Rosa26<sup>+/+</sup>; Col1a1<sup>T6B/T6B</sup>* females were crossed with *Rosa26rtTA/+; Col1a1<sup>T6B/T6B</sup>* males and doxycycline was administered by chow starting at 0.5 d.p.c. No viable pups positive for both the rtTA and T6B allele were observed (n = 15, p-value = 0.002, Fisher's exact test). (**B**) Pregnant females were kept on doxycycline diet from E13.5 to E18.5 and the pups delivered on E18.5 by c-section. Note the significantly smaller size of *Rosa26<sup>rtTA/+</sup>; Col1a1<sup>T6B/T6B</sup>* embryos relative to *Rosa26<sup>rtTA/rtTA</sup>;Col1a1<sup>+/+</sup>* control littermates. Lower row: YFP detection by epifluorescence in E18.5 pups of the indicated genotypes. (**C**) Comparison of intestine architecture in H&E sections from R26<sup>T6B</sup> and R26<sup>CTL</sup> mice (n = 3 for each genotype) maintained on doxycycline for 2 months. (**D**) Immunofluorescence imaging of the small intestine of R26<sup>T6B</sup> and R26<sup>CTL</sup> mice (n = 3–5 for each genotype) kept on doxycycline diet for a month (upper row), showing a reduction in lysozyme expression in Paneth cells in the crypts. Lysozyme expression in R26<sup>T6B</sup> mice returned to normal levels upon removal of doxycycline from the diet (lower row). (**E**) Peripheral blood analysis conducted in R26<sup>T6B</sup> and R26<sup>CTL</sup> mice (R26<sup>CTL</sup> n = 4; R26<sup>T6B</sup> n = 5). (**F**) Flow cytometric analysis of bone marrow of R26<sup>T6B</sup> and R26<sup>CTL</sup> mice kept on doxycycline diet for 3 weeks showing developmental block at the Pro-B to Pre-B. p-Values (from left to right): *p=0.0348, **p=0.0023, *p=0.0340, **p=0.0004, unpaired t-test. R26<sup>CTL</sup> n = 4; R26<sup>T6B</sup> n = 5. (**G**) Flow cytometry analysis of the bone marrow of control and R26<sup>T6B</sup> mice kept on doxycycline diet for 3 weeks. p-Values (from left to right): p=0.0994, **p=0.0092, **p=0.0085, *p=0.0312, unpaired t-test. R26<sup>CTL</sup> n = 4; R26<sup>T6B</sup> n = 5.

The online version of this article includes the following source data and figure supplement(s) for figure 3:

**Source data 1.** Complete blood counts (CBCs) of whole blood from R26T6B and R26CTL mice.

**Figure supplement 1.** Effects of FH-T6B-YFP fusion protein expression during development.

**Figure supplement 2.** Immunofluorescence imaging of the small and large intestine of R26<sup>T6B</sup> and R26<sup>CTL</sup> mice kept on doxycycline diet for a month.

**Figure supplement 3.** Sections from the colon and small intestine sections of R26<sup>T6B</sup> and control mice kept on doxycycline-containing diet for 2 months were probed by immunohistochemistry with an anti-Ki67 antibody.

**Figure supplement 4.** Detection of goblet cells by staining of acidic and neutral mucins in intestine sections from R26<sup>T6B</sup> and control mice kept on doxycycline diet for 2 months.

**Figure supplement 5.** Body weight of R26<sup>T6B</sup> (n = 5) and control (n = 8) female mice was assessed after 2-month administration of doxycycline-containing chow.

**Figure supplement 6.** Representative flow cytometry plots showing the gating strategy for the identification of hematopoietic stem and progenitor cells from whole bone marrow harvested from R26<sup>T6B</sup> and R26<sup>CTL</sup> mice maintained on doxycycline diet for 3 weeks.

**Figure supplement 7.** Representative flow cytometry plots showing the gating strategy for the identification of B cell lineage populations from whole bone marrow harvested from R26<sup>T6B</sup> and R26<sup>CTL</sup> mice maintained on doxycycline diet for 3 weeks.

the epithelium are maintained even in the absence of a functional miRISC. No significant change in the number of goblet cells was detected throughout the intestine (*Figure 3—figure supplement 4*), and mice maintained normal body mass throughout the period of doxycycline treatment (*Figure 3—figure supplement 5*), suggesting that general intestinal functions were not affected.

Although no obvious macroscopic, functional, or architectural abnormalities were caused by T6B expression in the intestine, we observed a reduction in lysozyme expression in Paneth cells in the crypts (*Figure 3D*, upper row). However, this phenotype was reversible as lysozyme signal in the crypts returned to normal levels when doxycycline was removed from the diet (*Figure 3D*, lower row), suggesting that T6B expression did not affect neither the viability of intestinal stem cells nor their self-renewal ability.

Complete blood counts showed a modest, but significant, decrease in erythrocytes volume (MCV) and hemoglobin content (MCH) in R26^T6B RBCs (*Figure 3E*, *Figure 3—source data 1*), analogously to what was reported in mice harboring targeted deletion of miR-451 (*Patrick et al., 2010*). Flow cytometric analysis of bone marrow showed a threefold depletion in Pre-B cells as well as a significant decrease in immature and mature circulating B cells in R26^T6B mice. We also observed a reciprocal increase in the frequency of Pro-B cells in the bone marrow of these animals (*Figure 3F*, *Figure 3—figure supplement 6*). These results are reminiscent of the partial block in B cell differentiation observed upon deletion of the miR-17–92 cluster (*Ventura et al., 2008*).

Further characterization of hematopoietic stem cells (HSCs) showed that the number of long-term repopulating hematopoietic stem cells (LT-HSC) was unaffected after 3 weeks of doxycycline exposure. However, we observed a modest decrease in short-term repopulating HSCs (ST-HSCs) and a concomitant increase in multipotent progenitors (MPPs) relative to controls (*Figure 3G*, *Figure 3—figure supplement 7*).

Collectively, these data suggest that in a subset of adult tissues miRISC function can be suppressed with minimal or no consequences on the ability of these tissues to maintain homeostasis.

## miRISC disruption impairs the regeneration of injured colon epithelium

Several studies have shown that the phenotype caused by targeted deletion of individual miRNAs often manifests only after the mutant animals are subjected to 'stress' (*Chivukula et al., 2014*; *Leung and Sharp, 2010*; *Mendell and Olson, 2012*; *van Rooij et al., 2007*). For example, ablation of miR-143/145 causes no apparent phenotype under homeostasis but severely impairs the ability of the mutant animals to respond to acute damage to the intestinal epithelium (*Chivukula et al., 2014*).

Prompted by these reports, and by our initial observation that prolonged T6B expression does not substantially affect intestinal homeostasis, we tested the consequences of miRISC disruption on the regenerating intestine. A cohort of R26^T6B and R26^CTL mice were kept on doxycycline-containing diet for 10 days, after which they were treated with dextran sulfate sodium (DSS), which induces severe colitis in mice (*Chivukula et al., 2014*; *Okayasu et al., 1990*).

A significant and progressive loss of body mass was observed in both groups during DSS treatment and 2 days following DSS removal (*Figure 4A*). However, R26^T6B mice lost body mass more rapidly than controls and reached critical health conditions 7 days after DSS removal. Three days after DSS removal, control animals started to regain weight, reaching the initial body mass within 5 days after DSS removal (*Figure 4A*). In contrast, R26^T6B mice failed to fully recover (*Figure 4A*), and all reached a humane endpoint within 5 days after DSS removal from the diet (*Figure 4B*).

Histological analysis confirmed that DSS treatment induced the disruption of the architecture of the epithelium and the appearance of ulcerative areas to a similar extent in both R26^T6B and R26^CTL control mice (*Figure 4C*, *Figure 4—figure supplement 1*). In contrast, although 5 days after DSS removal the integrity of the colonic epithelium of control mice was largely reestablished with the exception of isolated dysplastic areas (*Figure 4—figure supplement 2*), extensive ulcerated regions persisted in the colon of R26^T6B mice (*Figure 4C*). Importantly, we observed the presence of dysplastic epithelium in R26^T6B mice during and after DSS treatment, indicating that miRISC disruption does not completely abolish the potential of cells to proliferate, as also confirmed by Ki67 staining (*Figure 4D*). Therefore, we speculate that other factors, such as impaired stem cell maintenance or differentiation, may be responsible for the increased susceptibility of T6B-expressing colon to DSS treatment.

Chivukula and colleagues have shown that defective intestinal regeneration in the colon of miR-143/145-deficient mice is associated with upregulation of the miRNA-143 target IGFBP5 in the

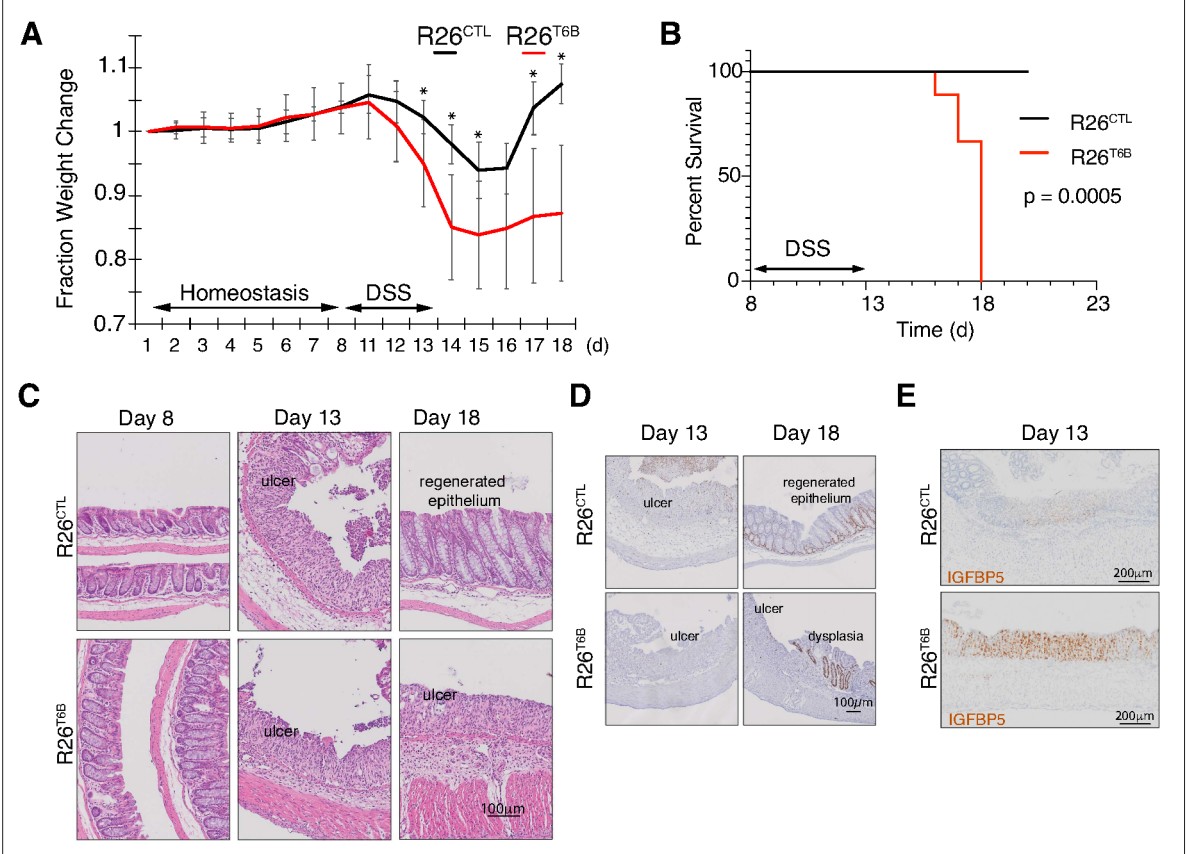

**Figure 4.** T6B-induced block of miRNA-induced silencing complex (miRISC) assembly leads to impaired intestinal regeneration. (**A**) R26^T6B and R26^CTL mice (n = 6 for each genotype) kept on doxycycline diet were treated with dextran sulfate sodium (DSS) for 5 days to induce inflammatory colitis and their weight was monitored daily. Data are presented as mean ± SD. p-Values (from left to right): *p=0.034, *p=0.005, *p=0.029, *p=0.024, *p=0.011, from unpaired t-test. (**B**) Kaplan–Meier curves of animals treated with DSS as described in panel (**A**). p-Value from log-rank test (**C**) Representative hematoxylin-eosin-stained sections of intestine of R26^T6B and R26^CTL mice (n = 3 for each genotype) at different time points pre- and post-DSS treatment. (**D**) Ki67 immunostaining of section of intestine at the indicated time points. (**E**) Sections from the large intestine of control and T6B mice euthanized at day 13 were subjected to RNA in situ hybridization with a probe against the IGFBP5 transcript. The results show increased levels of IGFBP5 mRNA in ulcerated areas of R26^T6B as compared to controls (n = 4 for each genotype).

The online version of this article includes the following figure supplement(s) for figure 4:

**Figure supplement 1.** Bar plots showing measurement of colon length, aggregated length of ulcers, percentage of colon with ulcers, area of ulcers, number of immune nodules, and the area of immune nodules performed on H&E longitudinal sections of colon from R26^CTL and R26^T6B mice 5 days post-dextran sulfate sodium (DSS) treatment.

**Figure supplement 2.** Representative immunohistochemistry image showing Ki67 signal in control mice (n = 3) 5 days after dextran sulfate sodium (DSS) treatment was discontinued.

mesenchymal compartment. The increased levels of IGFBP5 protein cause the inhibition of IGF1R signaling in the epithelium through a non-cell-autonomous mechanism, which ultimately prevented epithelial regeneration (*Chivukula et al., 2014*). Consistent with their findings, in situ hybridization analyses in the colon of DSS-treated R26^T6B mice showed a significant upregulation of IGFBP5 mRNA in the mesenchymal compartment compared to controls (*Figure 4E*). The extent of de-repression of IGFBP5 was comparable to that previously observed in miRNA-143/145 knockout mice (*Chivukula et al., 2014*), providing further evidence that T6B-mediated miRISC disassembly is an effective strategy to globally inhibit miRNA function in vivo.

Collectively, these results support a model whereby miRNA-mediated gene regulation, while dispensable to maintain normal colon homeostasis, becomes critical for its regeneration following acute damage.

## miRISC disruption impairs regeneration of the hematopoietic system

To further characterize the consequences of miRISC inhibition during tissue regeneration, we explored the possibility that other tissues may adopt a similar dynamic reliance on miRNA function.

Along with the intestinal epithelium, blood is one of the most rapidly turned over tissues in mice. HSCs reside as a predominantly quiescent population in the bone marrow and are rapidly induced to re-enter the cell cycle in response to external cues, such as infection or injury (*Ng and Alexander, 2017*). Furthermore, HSCs can be readily isolated by flow cytometry and transplanted, allowing the study of mechanisms underlying regeneration at the single-cell level.

To test the consequences of miRISC disruption in the regenerating hematopoietic system, we treated R26$^{T6B}$ and R26$^{CTL}$ mice on doxycycline-containing diet with a single dose of the cytotoxic drug 5-fluorouracil (5FU). 5-FU selectively depletes rapidly proliferating hematopoietic progenitors and leads to a compensatory increase in LT-HSC proliferation. Flow cytometry analysis of the bone marrow 7 days after 5FU-injection showed that T6B expression prevented this compensatory increase

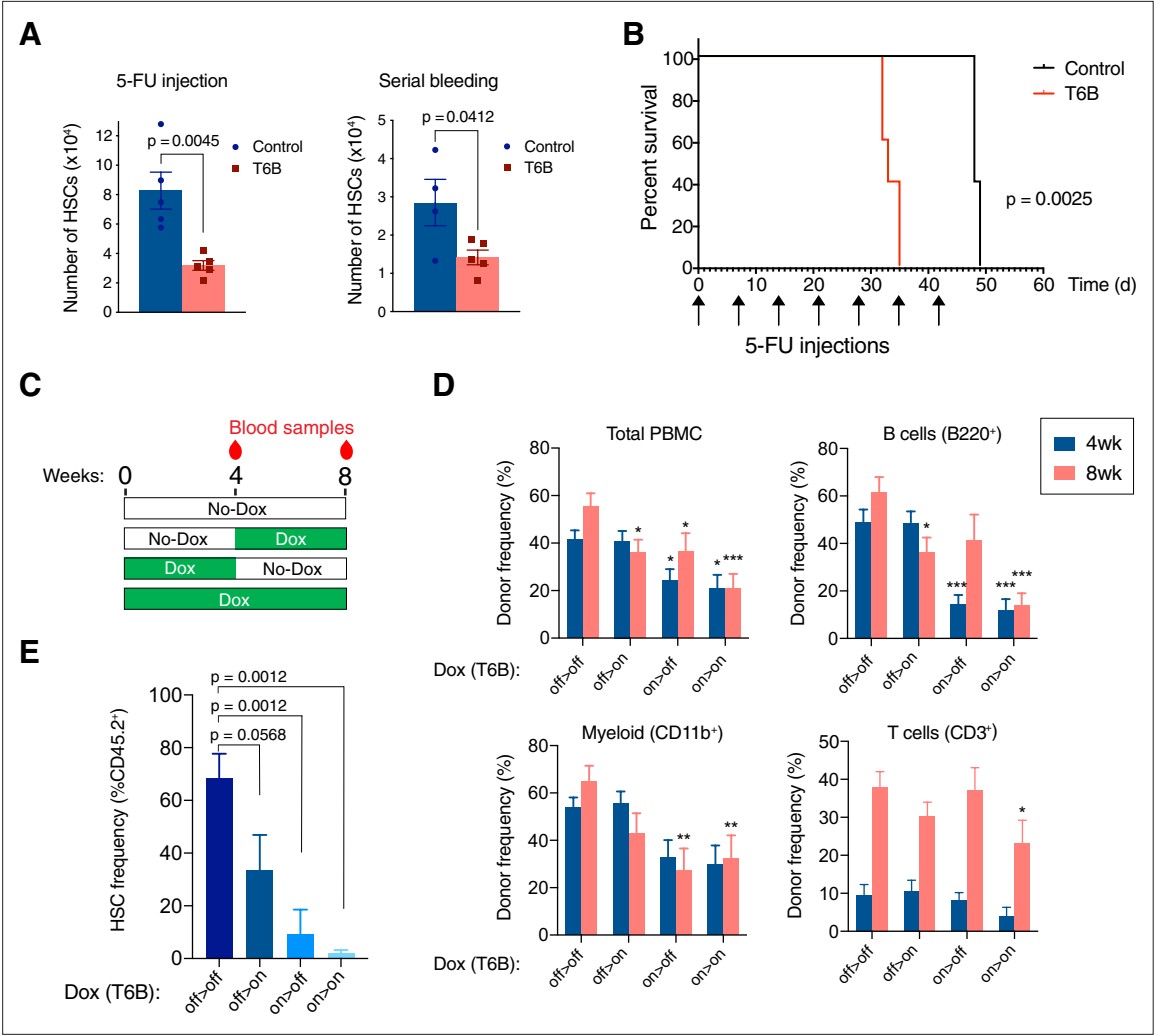

**Figure 5.** T6B-induced block of miRNA-induced silencing complex (miRISC) assembly impairs the regeneration of the hematopoietic system. (**A**) Long-term hematopoietic stem cell (HSC) in the bone marrow of R26$^{T6B}$ and R26$^{CTL}$ mice treated with 5-fluorouracil (5-FU) or subjected to repeated bleeding (n = 5 for each genotype). Mice were maintained on doxycycline-containing diet throughout the experiment. (**B**) Kaplan–Meier plots of R26$^{T6B}$ (n = 5) and R26$^{CTL}$ (n = 5) mice treated weekly with 5-FU for 7 weeks. (**C**) Schematic of the bone marrow transplantation experiments: T6B was induced at different time points post-transplantation, and multilineage reconstitution was assessed at the indicated time points by FACS. (**D**) FACS analysis conducted on the peripheral blood of irradiated recipients transplanted 1:1 with T6B-expressing and wild-type bone marrow, and maintained on doxycycline diet according to scheme shown in panel (**C**). Data are presented as mean ± SD. *p<0.05, **p<0.01, ***p<0.001, one-way ANOVA. off > off, n = 9; off > on, n = 10; on > off, n = 8; on > on, n = 8. (**E**) FACS analysis showing the frequency of T6B-extressing HSCs in the bone marrow of transplanted recipient mice kept on doxycycline diet according to scheme shown in panel (**C**). off > off, n = 5; off > on, n = 5; on > off, n = 4; on > on, n = 5, one-way ANOVA.

in LT-HSC. We observed an identical phenotype when R26^T6B and R26^CTL mice were bled repeatedly over a 3- week period to induce LT-HSC to re-enter the cell cycle (*Figure 5A*).

The decreased number of HSCs in the bone marrow of R26^T6B mice after a single 5-FU challenge compared to controls suggested that miRISC disruption impaired HSCs' ability to re-enter the cell cycle and regenerate the hematopoietic compartment. Consistent with this hypothesis, when injected with repetitive 5-FU doses, R26^T6B mice showed significantly shorter survival compared to controls (*Figure 5B*).

To measure the regenerative capacity of HSCs more directly in a context where T6B would only be expressed in hematopoietic cells, we performed competitive transplantation of T6B-expressing (CD45.2^+) and wild-type (CD45.1^+) bone marrows (1:1 ratio) into lethally irradiated hosts. The recipient animals were divided into four groups as shown in *Figure 5C*: (1) a control group that was never administered doxycycline; (2) a group maintained on a doxycycline-containing diet throughout the duration of the experiment (8 weeks); (3) a group treated with doxycycline starting 4 weeks after transplant; and (4) a group that was on doxycycline for only the first 4 weeks after transplant. Blood samples were taken at 4 and 8 weeks following the start of the experiment for analysis (*Figure 5C*). This experiment was designed to test the prediction that expression of T6B during the first 4 weeks following transplant, when the regenerative demand is highest and when we hypothesize miRNA-mediated gene repression is required, would more severely affect the ability of donor cells to contribute to the recipient hematopoietic reconstitution compared to T6B expression after homeostasis is reestablished.

Consistent with this prediction, mice that were administered doxycycline in the first 4 weeks post-transplant had significantly fewer CD45.2^+ peripheral blood mononuclear cells (PBMCs; *Figure 5D*). Contribution to the B cell population was particularly impaired by T6B expression, but this was reversed once the recipients were taken off of doxycycline, consistent with the developmental block described earlier (*Figure 3D*, *Figure 3—figure supplement 6*). Interestingly, the decrease in total CD45.2^+ PBMCs and CD45.2^+ myeloid cells was not reversed by doxycycline withdrawal, which suggested that the T6B-expressing CD45.2^+ HSCs might have been outcompeted by wild-type CD45.1^+ HSCs in these recipients (*Figure 5D*). Consistent with this hypothesis, we observed a significant reduction in CD45.2^+ HSCs only in the bone marrow of recipient animals that were fed a doxycycline-containing diet in the first 4 weeks post-transplant (*Figure 5E*).

Taken together, these results support a model where the miRNA-mediated gene regulation is conditionally essential for the maintenance of HSCs during acute regeneration but is largely dispensable under homeostasis.

## An essential role for miRNA-mediated gene repression in the skeletal muscle and in the heart

As previously discussed, we observed low or no expression of T6B in the heart and skeletal muscle of R26^T6B mice treated with doxycycline (*Figure 2—figure supplement 2*), consistent with previous reports indicating that rtTA expression from the endogenous *Rosa26* promoter is tissue restricted (*Premsrirut et al., 2011*). To extend the analysis of the phenotype caused by the loss of miRISC activity to these tissues, we crossed T6B transgenic mice with the *Rosa26-CAGs-rtTA3* strain (*Dow et al., 2014*) in which the modified chicken beta-actin with CMV-IE enhancer (CAG) promoter (*Niwa et al., 1991*) drives a more ubiquitous expression of the rtTA variant rtTA3 (hereafter CAG^T6B). As expected, the pattern and intensity of T6B expression upon dox administration in CAG^T6B mice and R26^T6B mice were largely overlapping, except for the heart and the skeletal muscle, for which significant T6B expression was only observed in CAG^T6B mice (*Figure 6A*, *Figure 2—figure supplement 2*). RNAseq analyses confirmed inhibition of miRNA function in both heart and skeletal muscle of CAG^T6B mice upon dox administration (*Figure 6B*).

In contrast to R26^T6B mice, CAG^T6B mice fed a doxycycline-containing diet showed a progressive decline in body mass (*Figure 6—figure supplement 1*) and died or reached a humane endpoint within 4–6 weeks (*Figure 6C*). The decrease in body mass was not caused by intestinal malabsorption as, similarly to what was observed in R26^T6B mice, we found no evidence of architectural defects throughout the intestine. In contrast, histopathological examination of heart and skeletal muscle showed severe alterations in both organs, including dilated cardiomyopathy and diffuse muscular degeneration (*Figure 6D*). All mice also showed necro-inflammatory changes in the liver, variable alterations in the pancreas, and increased urea nitrogen and alanine aminotransferase levels in the

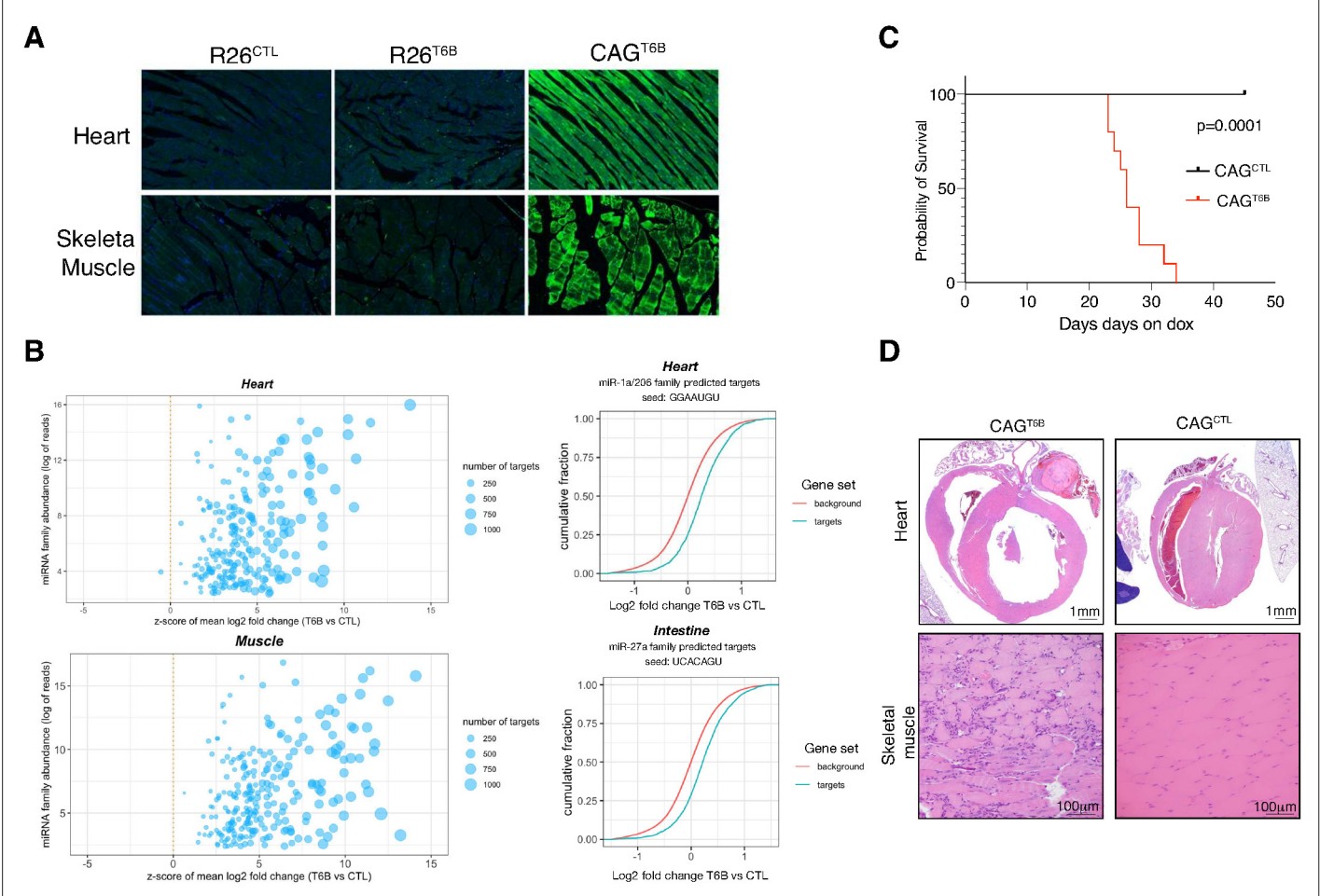

**Figure 6.** The microRNA (miRNA) pathway is essential in heart and skeletal muscle during homeostasis. (**A**) Detection of T6B expression with an anti-YFP antibody in the heart and skeletal muscle of R26^T6B, CAG^T6B, and R26^CTL mice maintained on doxycycline-containing diet for 7 days. (**B**) Total RNA extracted from the heart (upper panel) and the skeletal muscle (lower panel) of CAG^CTL and CAG^T6B mice (n = 3 for each strain) maintained on dox for 7 days was analyzed by RNAseq. Left panels: scatter plot showing the effect of T6B expression on targets of conserved miRNA families was generated as described in *Figure 1D*. The abundance of each miRNA family was calculated using dataset from *Isakova et al., 2020*. Right panels: representative cumulative distribution plot of log2-fold changes in expression of predicted targets of the indicated miRNA families. (**C**) Kaplan–Meier curves of CAG^T6B and CAG^CTL mice (n = 8 for each genotype) maintained on doxycycline throughout the duration of the experiment. p-Value from log-rank test. (**D**) Upper row: representative H&E staining showing marked dilation of the four cardiac chambers in hearts of CAG^T6B mice compared to controls (n = 9 for each genotype). Despite having thinner walls, the histomorphology of ventricular cardiomyofibers was within normal limits. Bottom row: representative H&E staining showing degenerative and regenerative changes in the skeletal muscle of the hind limbs of CAG^T6B mice compared to controls (n = 9 for each genotype).

The online version of this article includes the following figure supplement(s) for figure 6:

**Source data 1.** RNAseq, heart and muscle.

**Source data 2.** Z-scores and miRNA family abundance, heart and muscle.

**Figure supplement 1.** Body weight of CAG^T6B and control mice maintained on doxycycline for up to 45 days was assessed the day on which euthanasia was performed.

**Figure supplement 2.** Representative H&E staining showing vasculitis of the pulmonary veins as revealed by inflammatory immune cell infiltration of the vessel wall (arrows).

**Figure supplement 3.** T6B blocks miRNA activity in sea urchins and zebrafish.

serum. Such alterations are likely secondary to congestive heart failure and/or to severe muscle catabolism as they were not observed in R26^T6B mice. Another phenotype that distinguished the R26^T6B strain from the CAG^T6B strain was the presence in the latter of vasculitis of pulmonary veins (*Figure 6—figure supplement 2*). A likely explanation is that these lesions are caused by increased pressure in

the pulmonary veins secondary to congestive heart failure, but we cannot exclude that they reflect a direct effect of T6B expression on the pulmonary vasculature. Discriminating between these two possibility will require the use of transgenic mice harboring tissue-restricted rtTA transgenes.

The emergence of severe cardiac and skeletal muscle phenotypes, as opposed to the lack of obvious structural and functional abnormalities in most T6B-expressing tissues, points toward the existence of significant differences among adult tissues in their reliance on the miRNA pathway during homeostasis.

## Discussion

We report the generation of a novel genetically engineered mouse strain in which miRISC assembly and function can be temporally and spatially controlled in a reversible manner by a doxycycline-inducible transgene encoding a T6B-YFP fusion protein to address the role(s) miRNA-mediated gene regulation plays in vivo in adult tissues.

Surprisingly, in most adult tissues, we do not find an essential role for miRNA-mediated gene repression in organ homeostasis. A notable exception are the heart and the skeletal muscle, where miRISC inactivation in adult mice results in acute tissue degeneration and death even in the absence of tissue damage or exogenous stress.

Even though miRISC function is not overtly required for the homeostasis of other tissues, we have investigated the consequences of miRNA inhibition in the intestine and in the hematopoietic system of adult mice under homeostatic conditions and during tissue regeneration. These are tissues that periodically respond to external/internal stresses. In both tissues, we have found that miRISC activity is dispensable for homeostasis. However, miRNA function becomes essential during tissue regeneration following acute injury. These results lend experimental support to the hypothesis that a major role for miRNA-mediated gene repression is to support tissue adaptation to stress.

In previous studies where *Dicer1* was conditionally ablated in the skeletal muscle of adult mice, muscle regeneration was impaired after acute injury, but no effect on muscle morphology or function was observed during homeostasis (*Oikawa et al., 2019a*; *Oikawa et al., 2019b*; *Vechetti et al., 2019*). An explanation for this difference is that in the *Dicer1* conditional knockout experiments miRNA levels were only partially reduced even weeks after *Dicer1* ablation, likely reflecting the high stability of these short non-coding RNAs. The T6B mouse strain we describe here overcomes this major limitation and allows the rapid and effective inhibition of miRNA activity independently from the half-life of these molecules.

In this article, we have focused on the role of miRNA-mediated gene repression in adult mice. The same strategy for the acute inhibition of miRISC activity can in principle be applied to other organisms. We have found that expression of T6B in embryos of both sea urchin (*Paracentrotus lividus*) and zebrafish (*Danio rerio*) induces developmental defects and gene expression changes consistent with the essential role of the miRNA pathway during development (*Ambros and Horvitz, 1984*; *Chalfie et al., 1981*; *Lee et al., 1993*; *Reinhart et al., 2000*; *Song et al., 2012*; *Wienholds et al., 2003*; *Wightman et al., 1993*, *Figure 6—figure supplement 3*). Considering that in vitro T6B efficiently binds to AGO proteins from different non-mammalian organisms (*Hauptmann et al., 2015*), these findings are not unexpected, yet they highlight the usefulness of the T6B system for dissecting the miRNA pathway in a variety of animal models.

Despite its many advantages, the T6B mouse strain has also some unique limitations that need to be considered when designing and interpreting experiments.

First, although our biochemical and computational analysis of cells and tissues expressing T6B indicates that the peptide can effectively impair miRISC function, we cannot exclude some residual miRISC activity even in cells expressing high levels of the T6B transgene. The observation that we can recapitulate phenotypes observed in mice harboring complete targeted deletion of miR-143/145 miRNAs in the intestine (*Chivukula et al., 2014*) and of miR-17–92 and miR-451 in the hematopoietic system (*Koralov et al., 2008*; *Patrick et al., 2010*; *Ventura et al., 2008*) is reassuring in this respect. For example, consistent with observations made in the regenerating intestine of miRNA-143/145 knockout mice (*Chivukula et al., 2014*), we did not record any abnormalities or toxicity during the normal intestinal homeostasis of R26$^{T6B}$ mice, whereas T6B expression became lethal during intestinal regeneration. Moreover, in the hematopoietic system, abnormalities were mostly restricted to B cell maturation, which are consistent with a developmental block at the Pro-B to Pre-B transition found

in mir17–92 knockout mice (*Ventura et al., 2008*). Finally, we also observed a statistically significant decrease in hematocrit, erythrocyte volume, and hemoglobin content in adult T6B-expressing mice, analogous to what was reported in mice harboring targeted deletion of miR-451 (*Patrick et al., 2010*).

In contrast, some of our results markedly differ from the results obtained by conditional ablation of *Dicer1* in mice. For example, conditional knockout of *Dicer1* in the hematopoietic system has been reported to result in the rapid depletion of HSCs (*Guo et al., 2010b*). Furthermore, the lack of an overt phenotype in the intestine contrasts with previous reports showing that postnatal, conditional deletion of *Dicer1* results in depletion of goblet cells (*Biton et al., 2011*; *McKenna et al., 2010*), in addition to abnormal vacuolation and villous distortion in the small intestine (*Huang et al., 2012*; *McKenna et al., 2010*). We cannot exclude that these differences are due to an incomplete inactivation of the miRNA pathway in T6B mice, but an alternative explanation is that they reflect the well-characterized miRNA-independent functions of DICER.

Another limitation to be considered is the possibility that T6B expression impairs the activity of other complexes in addition to the miRISC. Although RNAseq analysis of cells expressing T6B has not revealed changes that are not explained by loss of miRNA-mediated gene repression and the phenotypes observed are consistent with loss of miRNA activity, this possibility cannot be formally excluded at this time. Further studies to experimentally identify T6B interactors in cells and tissues will be important to formally address this possibility.

In conclusion, we have developed a novel mouse strain that enables investigating the role of miRNA-mediated gene repression in adult organisms. The body of data presented here suggests that in adult animals miRNAs primarily provide for the ability to adaptively change gene expression in response to the physiological and pathological stresses that accompany metazoans' life. It is likely that the specific miRNAs and stresses differ based on the adult organ or tissue being studied, and the model we have generated will be useful to address these important aspects of miRNA biology.

## Materials and methods

**Key resources table**

| Reagent type (species) or resource | Designation | Source or reference | Identifiers | Additional information |
|---|---|---|---|---|
| Strain, strain background (*Mus musculus*) | T6B | This paper | Stock #036470 | The T6Bwt allele is integrated in the *Col1a1* locus |
| Strain, strain background (*M. musculus*) | CD45.1+ C57BL/6 (BoyJ) | Jackson Laboratory | RRID:IMSR_JAX:002014 | Carries the differential *Ptprc[a]* pan leukocyte marker |
| Strain, strain background (*M. musculus*) | C57BL/6J | Jackson Laboratory | RRID:IMSR_JAX:000664 | |
| Strain, strain background (*M. musculus*) | *Rosa26-CAGs-rtTA3* | Jackson Laboratory | RRID:IMSR_JAX:029627 | The CAG promoter drives the expression of rtTA3 |
| Cell line (*M. musculus*) | KH2 | PMID:16400644 | RRID:CVCL_C317 | Embryonic stem cells |
| Cell line (*M. musculus*) | DR4 | ATCC | RRID:CVCL_VK72 | Irradiated feeder cells |
| Transfected construct (*M. musculus*) | Silencer GAPDH siRNA | Thermo Fisher | #AM4624 | |
| Transfected construct (*M. musculus*) | Negative Control 1 siRNA | Thermo Fisher | #AM4611 | Nontargeting control |
| Antibody | Anti-E-cadherin (mouse monoclonal) | BD | #610181 | IF: (1:750) |
| Antibody | Anti-lysozyme (rabbit polyclonal) | Thermo Fisher | #RB-372-A1 | IF: (1:200) |
| Antibody | Anti-PH3 (mouse monoclonal) | Cell Signaling | #970 | IF: (1:200) |

*Continued on next page*

*Continued*

| Reagent type (species) or resource | Designation | Source or reference | Identifiers | Additional information |
|---|---|---|---|---|
| Antibody | Anti-YFP (rabbit polyclonal) | Invitrogen | #A11122 | IF: (1:250) |
| Antibody | Anti-Ki67 (rabbit monoclonal) | Cell Signaling | #12202 | IF: (1:400) |
| Antibody | Anti-Rabbit IgG, Alexa Fluor 488 (goat polyclonal) | Thermo Fisher | #A11034 | IF: (1:250) |
| Antibody | Anti-mouse IgG2a, Alexa Fluor 594 (goat polyclonal) | Thermo Fisher | #A-21135 | IF: (1:250) |
| Antibody | Anti-GFP (chicken polyclonal) | Abcam | #ab13970 | IF: (1:250) |
| Antibody | Rat IgG (rat polyclonal) | Sigma | #I-8015 | IF: (1:250) |
| Antibody | Anti-GW182 (rabbit polyclonal) | Bethyl | #A302-329A | WB: (1:1000, in 5% milk) |
| Antibody | Anti-Ago2 (rabbit monoclonal) | Cell Signaling | #2897 | WB: (1:1000) |
| Antibody | Anti-RPL26 (rabbit polyclonal) | Bethyl | #A300-686A | WB: (1:1000) |
| Antibody | Anti-GAPDH (mouse monoclonal) | Sigma | #G8795 | WB: (1:2000) |
| Antibody | anti-β-actin (mouse monoclonal) | Sigma | #A2228 | WB: (1:2000) |
| Antibody | Anti-tubulin (mouse monoclonal) | Sigma-Aldrich | #T9026 | WB: (1:2000) |
| Antibody | Anti-HA (rabbit monoclonal) | Cell Signaling | #C29F4 | WB: (1:1000) |
| Antibody | Anti-rabbit IgG, HRP-conjugated (donkey polyclonal) | GE Healthcare | #NA934 | WB: (1:10,000) |
| Antibody | Anti-mouse IgG, HRP-conjugated (sheep polyclonal) | GE Healthcare | #NA931 | WB: (1:10,000) |
| Antibody | Anti-AGO2 (mouse monoclonal) | WAKO | #011-22033 | IP: (1 µg/100 µl) |
| Antibody | Anti-AGO1-4 (mouse monoclonal) | EMD Millipore | #MABE56 | IP: (1 µg/100 µl) |
| Antibody | Anti-FLAG (mouse monoclonal) | Cell Signaling | #8146S | IP: (1 µg/100 µl) |
| Antibody | Anti-HA (mouse monoclonal) | Cell Signaling | #2367S | IP: (1 µg/100 µl) |
| Antibody | Anti-IgG1 isotype (mouse monoclonal) | Cell Signaling | #5415 | IP: (1 µg/100 µl) |
| Recombinant DNA reagent | pCAGGS-flpE-puro (plasmid) | Addgene | RRID:Addgene_20733 | Flippase recombinase-expressing vector |
| Recombinant DNA reagent | pgk-ATG-frt plasmid | Addgene | RRID:Addgene_20734 | |

*Continued on next page*

*Continued*

| Reagent type (species) or resource | Designation | Source or reference | Identifiers | Additional information |
|---|---|---|---|---|
| Sequence-based reagent | Col1a1 common _F | This paper | PCR primers | AATCATCCCAGGTG CACAGCATTGCGG |
| Sequence-based reagent | Col1a1 wildtype _R | This paper | PCR primers | CTTTGAGGGCTCAT GAACCTCCCAGG |
| Sequence-based reagent | Col1a1 mutant _R | This paper | PCR primers | ATCAAGGAAACCC TGGACTACTGCG |
| Sequence-based reagent | R26_F | This paper | PCR primers | AAAGTCGCTCT GAGTTGTTAT |
| Sequence-based reagent | R26a_R | This paper | PCR primers | GCGAAGAGTTTG TCCTCAACC |
| Sequence-based reagent | R26b_R | This paper | PCR primers | CCTCCAATTTTACACCTGTTC |
| Sequence-based reagent | T6B-YFP_F | This paper | PCR primers | GACTACAAGGACG ACGATGACAAG |
| Sequence-based reagent | T6B-YFP_R | This paper | PCR primers | GTTACTTGTACAG CTCGTCCATG |
| Commercial assay or kit | RNAscope 2.5 HD Detection Reagent, BROWN | ACD | #320771 | |
| Commercial assay or kit | RNAScope Igfbp5 Probe | ACD | #425738 | |
| Commercial assay or kit | Superose 6 10/300 GL | Cytiva | #GE17-5172-01 | Now available as Increase 10/300 GL, Cytiva #GE29-0915-96 |
| Commercial assay or kit | Novex NuPAGE SDS/ PAGE gel system | Thermo Fisher | #NP0321 | |
| Commercial assay or kit | EnVision + HRP | DAKO, Glostrup, Denmark | #K401111-2, RRID:AB_2827819 | |
| Commercial assay or kit | GFP-trap | Chromotek | #gtma-10 RRID:AB_2827592 | |
| Commercial assay or kit | TruSeq Stranded mRNA LT Kit, | Illumina | #RS-122-2102 | |
| Software, algorithm | OMERO | PMID:22373911 | RRID:SCR_002629 | |
| Software, algorithm | STAR v2.5.3a | PMID:23104886 | | |
| Software, algorithm | DESeq2 | PMID:25516281 | RRID:SCR_015687 | |
| Software, algorithm | miRbase version 21 | https://www.mirbase.org/ | | |
| Software, algorithm | TargetScan | PMID:26267216 | RRID:SCR_010845 | |
| Chemical compound, drug | Doxycyline-containing Rodent diet | Envigo | #TD01306 | 625 mg/kg doxycycline |
| Chemical compound, drug | Dextran sulfate sodium (DSS) | Cayman Chemical | #23250 | |
| Chemical compound, drug | Surgipath Decalcifier I | Leica Biosystems | #3800400 | Formic acid solution |
| Other | EDTA-free complete protease inhibitors | Sigma-Aldrich | #11836170001 | |
| Other | KnockOut DMEM | GIBCO | #10829018 | |

*Continued on next page*

*Continued*

| Reagent type (species) or resource | Designation | Source or reference | Identifiers | Additional information |
|---|---|---|---|---|
| Other | Phosphate inhibitors | Roche | #04906837001 | |
| Other | TRIzol Reagent | Thermo Fisher | #15596026 | |
| Other | DAPI stain | Sigma-Aldrich | #62248 | 5 μg/ml |
| Other | Mowiol 4-88 | Calbiochem | #475904100 GM | Mounting media |
| Other | GlutaMax | GIBCO | #35050061 | |
| Other | A/G PLUS-Agarose beads | Santa Cruz | #2003 | |
| Other | RIPA buffer | Sigma-Aldrich | #R0278 | |
| Other | Lipofectamine RNAiMAX | Thermo Fisher | #13778100 | Transfection reagent |
| Other | Alexa Fluor 488 tyramide signal amplification reagent | Life Technologies | B40953 | |

## Animal models

The *Rosa26$^{rtTA/rtTA}$*; *Col1a1$^{T6B/T6B}$* (R26$^{T6B}$) mice were generated by site-specific integration of the transgene coding for the FLAG-HA-T6B-YFP fusion protein within the *Col1a1* locus of KH2 embryonic stem cells (*Col1a1*-frt/*Rosa26 rtTA*; Beard et al., 2006). Briefly, the FLAG-HA-T6B-YFP (FH-T6B-YFP) DNA fragment was subcloned into the targeting vector, as described in 'Vectors and molecular cloning.' A mixture of 5 μg of the targeting vector and 2.5 μg of the pCAGGS-flpE-puro (Addgene #20733), Flippase recombinase-expressing vector was electroporated into KH2 cells, using 4D-Nucleofector core unit (Lonza), following the manufacturer's 'primary cells P3' protocol. Selection of targeted clones was initiated 48 hr after electroporation, using 150 μg hygromycin per ml of culture medium. 10 days later, individual hygromycin-resistant ES cell clones were analyzed by PCR to confirm correct integration of the knock-in allele. Clones carrying the correctly integrated knock-in allele were genotyped using a three-primer PCR, with the following primers: (1) 5′-AATCATCCCAGGTGCACAGCATTGCGG-3′; (2) 5′-CTTTGAGGGCTCATGAACCTCCCAGG-3′; and (3) 5′-ATCAAGGAAACCCTGGACTACTGCG-3′. A 287-bp-long PCR product indicates successful integration of the transgene into the *Col1a1* locus, while a 238-bp-long PCR product indicates a wild-type, untargeted locus. Two independent ES clones were injected into C57BL/6J albino blastocysts and backcrossed the resulting chimeras to C57BL/6J mice to achieve germline transmission of the recombinant allele. F1 animals were then intercrossed to generate animals expressing rtTA from the *Rosa26* locus under control of the *Rosa26* endogenous promoter, while expressing the T6B fusion protein from the *Col1a1* locus under control of the tetracycline-responsive element (TRE) and the minimal CMV promoter. Animals were genotyped as follows: to assess the presence of the transgene in the *Col1a1* locus, PCR was carried out as for the genotyping of KH2 cells. To assess the presence of the rtTA transgene in the *Rosa26* locus, a three-primer PCR was performed, with the following primers: (1) 5′-AAAGTCGCTCTGAGTTGTTAT-3′; (2) 5′-GCGAAGAGTTTGTCCTCAACC-3′; and (3) 5′-CCTCCAATTTTACACCTGTTC-3′. A 350-bp-long PCR product indicates the presence of the rtTA transgene into the *Rosa26* locus, while a 297-bp-long PCR product indicates the presence of a wild-type locus. *CAG$^{rtTA/rtTA}$*; *Col1a1$^{T6B/T6B}$* (CAG$^{T6B}$) mice were generated by backcrossing R26$^{T6B}$ mice with *Rosa26-CAGs-rtTA3* mice (a gift from Scott Lowe, MSKCC). In the *Rosa26-CAGs-rtTA3* mice, the knock-in allele has the CAG promoter driving the expression of the third-generation reverse tetracycline-regulated transactivator gene (rtTA3), all inserted into the *Gt(ROSA)26Sor* locus. In vivo doxycycline-dependent expression of the FLAG-HA-T6B-YFP transgene was achieved by feeding mice chow that contained doxycycline at the concentration of 625 mg/kg (Envigo #TD01306). Mice were maintained and euthanized in accordance with a protocol approved by the Memorial Sloan Kettering Cancer Center Institutional Animal Care and Use Committee. The T6B transgenic strain has been deposited at the Jackson Laboratory (JAX stock #036470).

## Necropsy, staining, and histopathology

Mice were euthanized with $CO_2$. Following gross examination, all organs were fixed in 10% neutral buffered formalin, followed by decalcification of bone in a formic acid solution (Surgipath Decalcifier

I, Leica Biosystems). Tissues were then processed in ethanol and xylene and embedded in paraffin in a Leica ASP6025 tissue processor. Paraffin blocks were sectioned at 5 µm, stained with hematoxylin and eosin (H&E), and examined by a board-certified veterinary pathologist. The following tissues were processed and examined: heart, thymus, lungs, liver, gallbladder, kidneys, pancreas, stomach, duodenum, jejunum, ileum, cecum, colon, lymph nodes (submandibular, mesenteric), salivary glands, skin (trunk and head), urinary bladder, uterus, cervix, vagina, ovaries, oviducts, adrenal glands, spleen, thyroid gland, esophagus, trachea, spinal cord, vertebrae, sternum, femur, tibia, stifle join, skeletal muscle, nerves, skull, nasal cavity, oral cavity, teeth, ears, eyes, pituitary gland, and brain. To detect goblet cells in the intestine, the AB/PAS kit (Thermo Fisher #87023) was used according to the manufacturer's instructions.

## Immunofluorescence

For the staining of intestine sections shown in *Figure 3* and *Figure 3—figure supplement 2*, formalin-fixed, paraffin-embedded (FFPE) slides were deparaffinized and rehydrated according to a standard xylene/ethanol series. After heat-induced epitope retrieval in sodium citrate (pH 6), tissue sections were permeabilized in triton X-100, blocked, and incubated with the following 1° antibodies: PH3 (Cell Signaling #970) at 1:200 dilution; lysozyme (Thermo Fisher #RB-372-A1) at 1:200 dilution; E-cadherin (BD #610181) at 1:750 dilution; YFP (Invitrogen #A11122) at 1:250 dilution; and Ki67 (Cell Signaling #12202) at 1:400 dilution. Next, cells were washed with PBS containing 0.05% Triton X and incubated with the following 2° antibodies: goat anti-rabbit IgG, Alexa Fluor 488 (Thermo Fisher #A11034) at 1:250 dilution; goat anti-mouse IgG2a, Alexa Fluor 594 (Thermo Fisher #A11029) at 1:250 dilution. For the staining of tissue sections shown in *Figures 2 and 4* and *Figure 2—figure supplement 2*, FFPE tissue sections were cut at 5 µm and heated at 58 °C for 1 hr. The antibody against GFP (Abcam, ab13970, 2 µg/ml) was incubated for 1 hr and detected with Leica Bond RX. Appropriate species-matched secondary antibody and Leica Bond Polymer anti-rabbit HRP were used, followed by Alexa Fluor 488 tyramide signal amplification reagent (Life Technologies, B40953). After staining, slides were washed in PBS and incubated in 5 µg/ml 4',6-diamidino-2-phenylindole (DAPI; Sigma-Aldrich) in PBS (Sigma-Aldrich) for 5 min, rinsed in PBS, and mounted in Mowiol 4-88 (Calbiochem). Slides were kept overnight at –20 °C before imaging.

## Immunohistochemistry

For immunohistochemistry, deparaffinized sections were subjected to antigen retrieval and processed with the EnVision + HRP kit (K401111-2, DAKO, Glostrup, Denmark) according to the manufacturer's instructions. A primary polyclonal antibody against Ki67 (Cell Signaling #12202) at 1:400 dilution was diluted in Antibody Diluent (DAKO #S0809) and incubated overnight at 4 °C. Next, sections were incubated in the provided anti-rabbit HRP-labeled polymer reagent, and detection was performed according to the manufacturer's protocol. Images were acquired using an Olympus BX-UCB slide scanner.

## RNA in situ hybridization

5 µm sections were obtained from FFPE colons from age/sex-matched mice. Before staining, tissue slides were deparaffinized, rehydrated, and permeabilized according to standard procedures. Detection was carried out using RNAscope 2.5 HD Detection Reagent, BROWN (ACD # 320771), with a specific RNAScope Igfbp5 Probe (ACD #425738), according to the manufacturer's instructions.

## Serum chemistry and hematology

For serum chemistry, blood was collected into tubes containing a serum separator, the tubes were centrifuged, and the serum was obtained for analysis. Serum chemistry was performed on a Beckman Coulter AU680 analyzer, and the concentration of the following analytes was determined: alkaline phosphatase, alanine aminotransferase, aspartate aminotransferase, creatine kinase, gamma-glutamyl transpeptidase, albumin, total protein, globulin, total bilirubin, blood urea nitrogen, creatinine, cholesterol, triglycerides, glucose, calcium, phosphorus, chloride, potassium, and sodium. Na/K ratio and albumin/globulin ratio were calculated. For hematology, blood was collected retro-orbitally into EDTA microtainers. Automated analysis was performed on an IDEXX Procyte DX hematology analyzer.

## DSS treatment and post-DSS treatment quantitative analyses

Mice kept in doxycycline-containing chow were treated for 5 days with 4% w/v DSS (FW 40.000; Cayman Chemical #23250) dissolved in drinking water. Body mass was monitored daily. Measurements of colon length, aggregated length of ulcers, percentage of colon with ulcers, area of ulcers, the number of immune nodules, and the area of immune nodules were obtained using OMERO (https://www.openmicroscopy.org/omero/). Measurements of these parameters were used to estimate the extent of damage and colitis induced by DSS treatment. All measurements were acquired from H&E-stained colon sections. Ulcer was defined as regions of colon with complete/partial loss of epithelial structure, accompanied by massive immune infiltrates. Colon length was measured by tracing the length of muscular layer of each colon. Length of ulcer was measured as the added length of each ulcerated region along the colon. Ulcer percentage was calculated as the length of ulcer/length of colon. The area of each individual ulcer was also measured and summed for each animal. Clear immune nodules are visible, showing aggregates of immune cells with high nucleus/cytoplasm ratio. Number and area of the immune nodules were summarized for each animal.

## Tissue isolation and total lysates preparation

Organs extracted from 8- to 12-week-old mice, perfused with PBS, were snap-frozen in liquid nitrogen and stored at −80 °C until further processing. To prepare total extract from solid tissues, tissues were pulverized using a mortar, resuspended in 1 ml of lysis buffer per $cm^3$ of tissue, and dounce-homogenized with a tight pestle until completely homogenized. Next, extracts were cleared by centrifugation at 20,000× g for 5 min followed by a second step of centrifugation at 20,000× g for 5 min. To prepare total extracts from cultured cells, pelleted cells were snap frozen in liquid nitrogen and stored at −80 °C until further processing. Pellets were then resuspended in lysis buffer, incubated for 10 min on ice, and cleared by centrifugation at 20,000× g. Two different lysis buffers were used, depending on the specific downstream application. For IP and SEC, lysates were prepared in SEC buffer (150 mM NaCl, 10 mM Tris-HCl pH 7.5, 2.5 mM $MgCl_2$, 0.01% Triton X-100). For western blotting applications, lysates were prepared in RIPA buffer (Sigma-Aldrich #R0278). Upon usage, both buffers were supplemented with the addition of EDTA-free complete protease inhibitors (Sigma-Aldrich #11836170001), phosphate inhibitors (Roche #04906837001), and 1 mM DTT.

## Cell lines and culture conditions

Cell lines were maintained in log-phase growth in a humidified incubator at 37 °C, 5% $CO_2$ prior to experimental manipulation. HCT116 colorectal adenocarcinoma cells were obtained from ATCC prior to this study and tested negative for Mycoplasma and were maintained in McCoy's medium supplemented with 10% heat-inactivated fetal calf serum (FCS, GIBCO, Cat#16141079), 10 U/ml penicillin/streptomycin, and 2 mM L-glutamine. MEFs were grown in Dulbecco's Modified Eagle Medium (DMEM) supplemented with 10% heat-inactivated FCS (GIBCO), 10 U/ml penicillin/streptomycin, and 2 mM L-glutamine. KH2 embryonic stem cells were cultured in gelatin-coated plates in the presence of irradiated DR4 Mouse Embryonic Fibroblasts (Thermo Fisher #A34966), and maintained in KnockOut DMEM (GIBCO, Cat#10829018), supplemented with 15% FCS (GIBCO), GlutaMax (GIBCO Cat#35050061), 100 µM non-essential amino acids (Sigma-Aldrich Cat#M7145), 1000 U/ml leukemia inhibitory factor (LIF, Millipore Cat#ESG1107), 10 U/ml penicillin/streptomycin (GIBCO Cat#15070063), 100 mM 2-mercaptoethanol (Bio-Rad Cat#1610710), and nucleosides (Millipore Cat#ES-008-D).

## Flow cytometry

Analysis of bone marrow populations was performed by harvesting femurs and tibiae from euthanized mice. Bone marrow was isolated by centrifugation, resuspended in FACS buffer (PBS with 2% FCS), and passed through a 40 µm cell strainer to make a single-cell suspension. Nonspecific antibody binding was blocked by incubation with 10 µg/ml rat IgG (Sigma #I-8015) for 15 min on ice. Antibodies used to identify HSCs included a cocktail of biotinylated lineage antibodies (Gr1, CD11b, TER119, B220, CD3, CD4, CD8), CD117 (c-kit) APC (2B8), Sca-1 (D7) PE-cy7, CD150 PE, and CD48 Pacific Blue. B cell progenitors were identified with the following antibodies: B220, CD19, CD25, CD43, IgM, IgD, and c-kit. For analysis of PBMCs, blood was collected retro-orbitally from live mice into EDTA microtainers. Whole blood was lysed in ACK buffer for 5 min at room temperature, washed with FACS buffer, and pelleted prior to antibody staining. Mature blood populations were identified with

the following antibodies: CD45.1, CD45.2, Gr1, CD11b, B220, and CD3. Cells were incubated with primary antibodies for 45 min, washed once with FACS buffer, and incubated with BV711 streptavidin conjugate for 15 min. All incubations were carried out on ice and protected from light. Antibodies were purchased from BioLegend or eBioscience.

## Bone marrow transplantation

8- to 12 -week-old CD45.1[+] C57BL/6 (BoyJ) mice (JAX) were lethally irradiated by exposure to 1100 cGy of gamma irradiation from a cesium source, administered in two doses, split 4 hr apart. Bone marrow suspensions from CAG[T6B] (CD45.2[+]) and BoyJ mice were counted, mixed 1:1, and transferred intravenously by retro-orbital injection into isoflurane-anesthetized, irradiated recipients.

## Size-exclusion chromatography (SEC)

SEC was performed using a Superose 6 10/300 GL prepacked column (GE Healthcare) equilibrated with SEC buffer essentially as previously described (*La Rocca et al., 2015*; *Olejniczak et al., 2013*). Briefly, 400 µl (1.5–2 mg) of total extracts precleared by centrifugation were run on the SEC column at a flow rate of 0.3 ml/min. 1 ml fractions were collected. Proteins were extracted from each fraction by TCA precipitation following standard procedures and run on SDS-PAGE gels for western blotting analysis.

## Western blotting and antibodies

Western blotting was performed using the Novex NuPAGE SDS/PAGE gel system (Invitrogen). Total cell lysates were run either on 3–8% Tris-acetate or 4–12% Bis-Tris precast gels, transferred to nitrocellulose membranes, and probed with antibodies specific to proteins of interest. Detection and quantification of blots were performed on Amersham hyperfilm ECL (Cytiva #28906839) and developed on film processor SRX-101A (Konica). Antibodies used for western blots were obtained from commercial sources as follows: anti-GW182 (Bethyl #A302-239A), anti-Ago2 (Cell Signaling #2897), anti-PABP1 (Cell Signaling #4992), anti-RPL26 (Bethyl #A300-686A), anti-GAPDH (Sigma #G8795), anti-β-actin (Sigma #A2228), anti-GFP (Roche #11814460001), anti-tubulin (Sigma-Aldrich #T9026), anti-HA (Cell Signaling #C29F4), anti-rabbit IgG, HRP-conjugated (GE Healthcare #NA934), and anti-mouse IgG, HRP-conjugated (GE Healthcare #NA931).

## Immunoprecipitation (IP)

For IP of AGO-T6B complexes from human HCT116 cells, 500 µg of lysates in 500 µl of SEC buffer were incubated for 3 hr with primary antibodies directed to either AGO proteins (WAKO anti-AGO2 #011-22033, EMD Millipore anti-panAGO #MABE56) or directed to T6B-fusion protein (Cell Signaling anti-FLAG #8146S , Cell Signaling anti-HA #2367S ) or mouse IgG1 isotype control (Cell Signaling #5415). Next, lysates were incubated with 20 µl of protein A/G PLUS-Agarose beads (Santa Cruz #2003) for 1 hr. For IP of AGO-T6B complexes from mouse tissues, 500 µg of lysates in 500 µl of SEC buffer were incubated for 2 hr with GFP-trap magnetic agarose beads (Chromotek #gtma-10) or binding control beads (Chromotek #bmab-20). The immune complexes were run on SDS-PAGE and analyzed by western blotting.

## Vectors and molecular cloning

The targeting vector expressing the FH-T6B-YFP under control of TRE and CMV minimal promoter was generated from a modified version of the pgk-ATG-frt plasmid (Addgene plasmid #20734), in which the region of pgk-ATG-frt comprised between the EcoRI site and the PciI site was substituted with the rabbit β-globin polyadenylation signal (RBG pA). The FH-T6B-YFP DNA insert was generated by PCR using the plasmid pIRES-Neo-FH-T6B-YFP[58] as a template. PCR was carried out using the following primers: forward: 5′-GACTACAAGGACGACGATGACAAG-3′, reverse: GTTACTTGTACA GCTCGTCCATG. Next, the modified pgk-ATG-frt was cut with NcoI, filled-in to produce blunt ends, dephosphorylated, and ligated to the PCR-generated FH-T6B-YFP DNA fragment according to standard subcloning procedures. A scheme of the cloning strategy is shown as follows:

**Scheme 1.** Cloning strategy for the generation of the targeting vector expressing the FH-T6B-YFP transgene.

To generated cell lines expressing either FH-T6B-YFP or FH-T6B$^{Mut}$-YFP fusion proteins in a doxycycline-inducible manner, a modified version of the retroviral vector pSIN-TREtight-HA-UbiC-rtTA3-IRES-Hygro (hereafter TURN vector, a gift from Scott Lowe) was used to transduce commercially available HCT116 and MEFs cell lines. TURN is an all-in-one Tet-on vector that includes (1) the rtTA3 gene under the human ubiquitin C promoter and (2) the transgene of interest driven by a TRE/CMV promoter. We used the pIRES-Neo-FH-T6B-YFP described in *Hauptmann et al., 2015* as a template to generate by PCR the DNA fragments coding either for FH-T6B-YFP or for FH-T6BMut-YFP fusion proteins. DNA fragments were then inserted into the XhoI/EcoRI-digested TURN vector to generate TURN$^{T6B}$ and TURN$^{T6Bmut}$ vectors used for the transduction of parental HCT116 and MEFs.

## Small RNA transfection

Silencer GAPDH siRNA (Thermo Fisher AM4624) and Silencer Select Negative Control 1 siRNA (Thermo Fisher AM4611) small RNAs were transfected at 10 pM per $1 \times 10^6$ cells. MEFs were reverse transfected using Lipofectamine RNAiMAX. Lipofectamine RNAiMAX was combined with 20 µM small RNAs at a 4:3 ratio (vol:vol) in Opti-MEM and incubated for 20 min at room temperature. Trypsinized cells were added to culture dishes containing siRNAs and Lipofectamine RNAiMAX at $3.8 \times 10^4$ cells per cm$^2$. Three volumes of complete medium were added to culture dishes, and cells were incubated for 2–3 days before further processing.

## Small RNA sequencing

Total RNA was extracted from MEFs transduced with the retroviral vectors encoding a doxycycline-inducible T6B or T6B$^{mut}$ transgene and cultured in the presence or absence of doxycycline. Small RNA-seq library preparation was as described in *Hafner et al., 2011*. Briefly, 1 µg total RNA was ligated to nine distinct pre-adenylated 26-nt 3′-adapters with a 5-nt barcode using a mutated and truncated Rnl2 followed by urea gel purification and size selection and 5′-adapter ligation with Rnl1. This ligation reaction was again gel-purified and size-selected for fully ligated product and reverse-transcribed using SuperScript III RT followed by PCR amplification using Taq polymerase for 25 cycles. The final PCR product was separated on a 2% agarose gel in TBE buffer and extracted using the QIAgen gel extraction kit according to the manufacturer's instructions including all optional steps. After high-throughput sequencing, small RNA reads were aligned to a miRNA genome index built from 1915 murine pre-miRNA sequences from miRbase version 21 (*Kozomara et al., 2019*; ftp://mirbase.org/pub/mirbase/21/) using Bowtie v2.4.296. Mature miRNA abundance was calculated by counting reads falling within 4 bps at each of the 5′ and 3′ end of the annotated mature miRNAs. miRNA seed family data were downloaded from the TargetScan website at http://www.targetscan.org/cgi-bin/targetscan/data_download.cgi?db=mmu_71. For miRNA family-level analysis, read counts mapping to members of the same miRNA family were summed up.

## RNAseq analysis

Total RNA from heart, skeletal muscle, colon, and liver of sex-matched littermate animals, and total RNA from cell lines were extracted using TRIzol Reagent (Invitrogen) according to the manufacturer's instructions and subjected to DNase (QIAGEN) treatment. After RiboGreen quantification and quality control by Agilent BioAnalyzer, 500 ng of total RNA with RIN values of 7.0–10 underwent polyA

selection and TruSeq library preparation according to the instructions provided by Illumina (TruSeq Stranded mRNA LT Kit, Cat#RS-122-2102), with eight cycles of PCR. Samples were barcoded and run on a HiSeq 4000 in a PE50/50 run using the HiSeq 3000/4000 SBS Kit (Illumina). An average of 34 million paired reads was generated per sample. The percent of mRNA bases averaged 60% over all samples. Reads were aligned to the standard mouse genome (mm10) using STAR v2.5.3a (*Dobin et al., 2013*). RNA reads aligned were counted at each gene locus. Expressed genes were subjected to differential gene expression analysis using DESeq2 (*Love et al., 2014*), and log2-fold changes were determined comparing T6B-expressing tissues to controls.

## Z-score calculation

For each conserved miRNA families, the mean log2-fold change of predicted targets, as defined by TargetScan, compared to the rest of the transcriptome (background), was calculated. The means were converted to z-scores as described by *Kim and Volsky, 2005*: Z-score = $(Sm - \mu)*m^{1/2}*\partial^{-1}$, where $Sm$ is the mean of log2-fold changes of genes for a given gene set, $m$ is the size of the gene set, and $\mu$ and $\partial$ are the mean and the standard deviation of background log2-fold change values.

## Real-time quantitative PCR

Real-time quantitative PCR analysis to assess the expression levels of the territorial marker genes involved in the developmental gene regulatory network of the sea urchin was conducted as previously described by *Cavalieri et al., 2009*. Briefly, total RNA from batches of 150 microinjected embryos was extracted by using the High Pure RNA Isolation kit (Roche). RNA samples were treated with reagents provided by the Turbo DNA-free kit (Ambion) and resuspended in a final volume of 30 μl. Reverse transcription into cDNA was performed in an 80 μl reaction using random hexamers and the TaqMan Reverse Transcription Reagents kit (Applied Biosystems). The resulting cDNA sample was further diluted, and the equivalent amount corresponding to one embryo was used as template for Q-PCR analysis. Q-PCR experiments were performed from two different batches, and all reactions were run in triplicate on the 7300 Real-Time PCR system (Applied Biosystems) using SYBR Green detection chemistry (Applied Biosystems). ROX was used as a measure of background fluorescence, and *MBF-1* and *z12* mRNAs were used as internal controls. At the end of the amplification reactions, a 'melting-curve analysis' was run to confirm the homogeneity of all Q-PCR products. Calculations from Q-PCR raw data were performed by the RQ Study software version 1.2.3 (Applied Biosystems) using the comparative Ct method (Ct). Oligonucleotide primer pairs used for qPCR reactions and amplicon lengths have been described previously (*Cavalieri et al., 2008*, *Cavalieri et al., 2011*, *Cavalieri and Spinelli, 2014*, *Cavalieri et al., 2017*, *Turturici et al., 2018*).

## Acknowledgements

This work was funded by the Starr Foundation's Tri-Institutional Stem Cell Initiative (AV, TL, DB, and TT) and by the NIH/NCI (grants R01CA149707 and R01CA245507 to AV and P30 CA008748 to CBT). YM was supported by a Medical Scientist Training Program grant from the National Institute of General Medical Sciences of the National Institutes of Health under award number: T32GM007739 to the Weill Cornell/Rockefeller/Sloan Kettering Tri-Institutional MD-PhD Program. We acknowledge the use of the following core facilities at the Memorial Sloan Kettering Cancer Center (MSKCC): The Molecular Cytology Core; The Mouse Genetics Core Facility; The Laboratory of Comparative Pathology, and the Integrated Genomics Operation Core, funded by the NCI Cancer Center Support Grant (CCSG, P30 CA08748), Cycle for Survival, and the Marie-Josée and Henry R Kravis Center for Molecular Oncology. We thank Sebastien Monette and Ileana Miranda for their contribution in the phenotypic analysis of R26^T6B and CAG^T6B mouse strains; Davide Pradella, Rui Gao, Saurabh Yadav, and members of the Benezra laboratory for discussion and suggestions. Finally, we thank Jaqueline Candelier for the handling and processing of tissue samples used in the phenotypic analyses of the R26^T6B and CAG^T6B mouse strains.

## Additional information

### Competing interests

Craig B Thompson: is a founder of Agios Pharmaceuticals and a member of its scientific advisory board. He is also a former member of the Board of Directors and stockholder of Merck and Charles River Laboratories. He is a named inventor on patents related to cellular metabolism. Potentially relevant patents on which he is a named inventor include the following: (i) L-2-hydroxyglutarate and stress induced metabolism (United States Patent #10,450,596). (ii) Single diastereomers of 4-fluoroglutamine and methods of their preparation and use (United States Patent #8,747,809). A complete list of patents can be found at the following link: https://tinyurl.com/y35qvajq. The other authors declare that no competing interests exist.

### Funding

| Funder | Grant reference number | Author |
|---|---|---|
| National Cancer Institute | R01CA149707 | Andrea Ventura |
| National Cancer Institute | R01CA245507 | Andrea Ventura |
| National Cancer Institute | P30 CA008748 | Craig B Thompson |
| Starr Foundation | | Gaspare La Rocca Tullia Lindsten Andrea Ventura |
| National Institute of General Medical Sciences | T32GM007739 | Yilun Ma |
| National Cancer Institute | P30 CA08748 | Gaspare La Rocca |

The funders had no role in study design, data collection and interpretation, or the decision to submit the work for publication.

### Author contributions

Gaspare La Rocca, Conceptualization, Data curation, Formal analysis, Funding acquisition, Investigation, Methodology, Supervision, Visualization, Writing – original draft, Writing – review and editing; Bryan King, Bing Shui, Conceptualization, Formal analysis, Investigation, Methodology, Visualization, Writing – original draft, Writing – review and editing; Xiaoyi Li, Data curation, Formal analysis, Investigation, Software, Visualization, Writing – original draft, Writing – review and editing; Minsi Zhang, Formal analysis, Investigation, Methodology, Writing – review and editing; Kemal M Akat, Data curation, Investigation, Writing – review and editing; Paul Ogrodowski, Chiara Mastroleo, Kevin Chen, Investigation, Resources; Vincenzo Cavalieri, Investigation, Methodology, Visualization, Writing – review and editing; Yilun Ma, Investigation; Viviana Anelli, Investigation, Methodology; Doron Betel, Data curation, Formal analysis, Funding acquisition, Investigation, Writing – review and editing; Joana Vidigal, Gunter Meister, Conceptualization, Investigation, Methodology, Writing – review and editing; Thomas Tuschl, Conceptualization, Data curation, Formal analysis, Project administration, Writing – review and editing; Craig B Thompson, Tullia Lindsten, Conceptualization, Funding acquisition, Supervision, Writing – review and editing; Kevin Haigis, Conceptualization, Formal analysis, Investigation, Methodology, Supervision, Visualization, Writing – review and editing; Andrea Ventura, Conceptualization, Data curation, Formal analysis, Funding acquisition, Investigation, Methodology, Project administration, Software, Supervision, Visualization, Writing – original draft, Writing – review and editing

### Author ORCIDs

Gaspare La Rocca (iD) http://orcid.org/0000-0003-1277-0566
Bing Shui (iD) http://orcid.org/0000-0002-5956-130X
Kemal M Akat (iD) http://orcid.org/0000-0002-9012-3551
Kevin Chen (iD) http://orcid.org/0000-0002-0674-1411
Gunter Meister (iD) http://orcid.org/0000-0002-2098-9923
Craig B Thompson (iD) http://orcid.org/0000-0003-3580-2751
Andrea Ventura (iD) http://orcid.org/0000-0003-4320-9907

### Ethics

This study was performed in strict accordance with the recommendations in the Guide for the Care and Use of Laboratory Animals of the National Institutes of Health. All of the animals were handled according to approved institutional animal care and use committee (IACUC) protocols (#10-10-022) of Memorial Sloan Kettering Cancer Center.

### Decision letter and Author response

Decision letter https://doi.org/10.7554/eLife.70948.sa1
Author response https://doi.org/10.7554/eLife.70948.sa2

## Additional files

### Supplementary files

• Transparent reporting form

### Data availability

Processed sequencing data are included as source data. Fastq files have been deposited to GEO (GEO accession number: GSE179588).

The following dataset was generated:

| Author(s) | Year | Dataset title | Dataset URL | Database and Identifier |
|---|---|---|---|---|
| Ventura A, Li X, La Rocca G, King B | 2021 | Inducible and reversible inhibition of miRNA-mediated gene repression in vivo | https://www.ncbi.nlm.nih.gov/geo/query/acc.cgi?acc=GSE179588 | NCBI Gene Expression Omnibus, GSE179588 |

The following previously published datasets were used:

| Author(s) | Year | Dataset title | Dataset URL | Database and Identifier |
|---|---|---|---|---|
| Fehlmann I, Quake K | 2020 | A mouse tissue atlas of small noncoding RNA | https://www.ncbi.nlm.nih.gov/geo/query/acc.cgi?acc=GSE119661 | NCBI Gene Expression Omnibus, GSE119661 |

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
