## [Decision Letter]

**Acceptance summary:**

In this manuscript, the authors have extensively characterized a new tool for studying the phenotypic roles of miRNA-mediated regulation in mice. Analysis of this model has convincingly demonstrated that miRNA activity is dispensable for homeostasis in most adult tissues, with the notable exception of heart and skeletal muscle. This work provides an extremely useful tool for the study of miRNAs in vivo and provides new insights into the roles of miRNAs in adult mammalian tissues.

**Decision letter after peer review:**

Thank you for submitting your article "Inducible and reversible inhibition of miRNA-mediated gene repression in vivo" for consideration by *eLife*. Your article has been reviewed by 3 peer reviewers, including Ashish Lal as the Reviewing Editor and Reviewer #1, and the evaluation has been overseen by Kevin Struhl as the Senior Editor. The following individual involved in review of your submission has agreed to reveal their identity: Joshua T Mendell (Reviewer #2).

Essential revisions:

In this manuscript the authors developed an inducible and reversible system for inhibition of miRNA-mediated gene repression in vivo. The authors demonstrate the use of this system by investigating the consequences of inhibition of miRNA activity in adult animals. Interestingly, they found that global loss if miRNA function has different effects on different tissues and organs and in some tissues the effect is context dependent. Overall, this work is novel, interesting and data presented is solid. The results are presented in a clear and straightforward way. While most aspects of this work are very intriguing and supported well by solid experiments, a few additional analyses would provide additional information that would be interesting to readers.

1. An overall histopathologic analysis of P0 T6B mice exposed to dox between E13.5-E18.5 could be interesting to see if the development of any tissues are particularly sensitive to inhibition of miRNA-mediated silencing during this window. This would be straightforward if fixed P0 mice are available, although this analysis could be the subject of a future study if tissues are not readily available.

2. Figure 2 – supplement 2 shows T6B activation in lung in CAG-T6B but not R26-T6B animals. Does activation of the transgene in CAG-T6B mice result in any noticeable abnormalities in adult lung?

---

## [Author Response]

Essential revisions:1. An overall histopathologic analysis of P0 T6B mice exposed to dox between E13.5-E18.5 could be interesting to see if the development of any tissues are particularly sensitive to inhibition of miRNA-mediated silencing during this window. This would be straightforward if fixed P0 mice are available, although this analysis could be the subject of a future study if tissues are not readily available.

We agree with the Reviewer that a detailed characterization of the consequences of acute miRNA inhibition on mouse development is of great interest. We have asked the MSKCC mouse pathology service to carry out a pathologic analysis of the embryos described in Figure 3B and Figure 3—figure supplement 1. The results of this analysis confirmed the observed general delay in development of the T6B mice compared to wild type controls, but did not highlight any specific developmental defect that can be detected by histologic analysis of H and E stained sections.

We have initiated a series of more detailed studies, comparing different timing of T6B induction during embryonic development, and examining the effect of miRNA inhibition on in vitro differentiation trajectories by using single cell RNAseq analysis. We are thrilled about these studies, but they will require considerable time and effort and therefore, if the Editor and the Reviewers agree, we would prefer to present them in a follow-up manuscript.

2. Figure 2 – supplement 2 shows T6B activation in lung in CAG-T6B but not R26-T6B animals. Does activation of the transgene in CAG-T6B mice result in any noticeable abnormalities in adult lung?

We thank the Editor and Reviewers for the opportunity to clarify this aspect of the phenotypic analysis of the CAG-T6B strain that had been superficially described in our previous version of the manuscript. To address this point, we asked an experienced mouse pathologist from the MSKCC animal core to review the lung histology of adult CAG-T6B mice. As described in the revised manuscript, this analysis revealed marked vasculitis of the pulmonary veins (Figure 6—figure supplement 2). The most likely explanation for this phenotype is that these lesions are secondary to the increased pulmonary vein pressure caused by congestive heart failure, but we cannot exclude that a primary effect caused by T6B expression in the lung contributed to this phenotype. Follow-up studies using transgenic animals harboring tissue-restricted expression of rtTA will be needed to conclusively address this important point.